# Interactions between the protein barnase and co-solutes studied by NMR
Clare R. Trevitt [1,2], D. R. Yashwanth Kumar [1], Nicholas J. Fowler [1] & Mike P. Williamson [1] ✉

Protein solubility and stability depend on the co-solutes present. There is little theoretical basis for selection of suitable co-solutes. Some guidance is provided by the Hofmeister series, an empirical ordering of anions according to their effect on solubility and stability; and by osmolytes, which are small organic molecules produced by cells to allow them to function in stressful environments. Here, NMR titrations of the protein barnase with Hofmeister anions and osmolytes are used to measure and locate binding, and thus to separate binding and bulk solvent effects. We describe a rationalisation of Hofmeister (and inverse Hofmeister) effects, which is similar to the traditional chaotrope/kosmotrope idea but based on solvent fluctuation rather than water withdrawal, and characterise how co-solutes affect protein stability and solubility, based on solvent fluctuations. This provides a coherent explanation for solute effects, and points towards a more rational basis for choice of excipients.

The majority of therapeutic drugs by value are "biologics", in other words proteins. By comparison to traditional small-molecule compounds, biologics have the virtue of being much more specific, but they are also much more expensive, and have a much shorter shelf life because they need to be made up in solution, and maintained in solution over the lifetime of the formulation. There is thus intense interest in pharmaceutical companies in ways to stabilize proteins and keep them in solution, and functional, for as long as possible. Despite this sustained interest over many years, so far there is no clear theoretical framework[1–4], and essentially formulation consists of trying out previously successful ideas[5]. The aim of this work is to establish some of the fundamental science, in particular the relative importance of binding and solvation, in order to produce a more coherent framework for understanding the effects of co-solutes on proteins.

There are a large number of theories to account for the effects of co-solutes (typically described as excipients in formulation science[6]) on proteins. Many appear different and mutually inconsistent. To some extent, this is genuinely true, and to some extent it is only apparent because any biophysical theory can usually be expressed in multiple ways, in completely different languages. Thus, part of the problem addressed here is to use the most meaningful and helpful language.

Hofmeister described the effects of ions on protein solubility and stability in the 1880s and 1890s. He found that the effect of cations is weak, while anions have a stronger effect. The anion series is often written as

$$PO_4^{3-} > SO_4^{2-} > HPO_4^{2-} > F^- > Cl^- > NO_3^- > Br^- > ClO_3^- > I^- > ClO_4^- > SCN^-$$

| Increase in protein stability | Decrease in protein stability |
| Decrease in protein solubility | Increase in protein solubility |

in which small well-solvated "hard" ions with high charge density (phosphate, sulfate) act to increase protein stability but decrease protein solubility, while at the other end larger less-solvated ions with more diffuse charge density (eg thiocyanate, perchlorate) have the opposite effect, in that they decrease stability and increase solubility. This series can be applied to a wide range of proteins and affects other properties too, including membrane interactions, colloidal charge, and surface tension[7–9]. The standard explanation for this phenomenon, which became popular from the 1930s[10], is essentially structural: it describes high charge density ions at the left of the series as kosmotropes (water "structure makers"), and low charge density ions at the right as chaotropes ("structure breakers"). Kosmotropes are suggested to order water molecules well beyond their direct hydration sphere, and chaotropes to disorder water[11]. Kosmotropes are therefore described as removing water from the protein surface, leading to a salting-out effect, whereas chaotropes allow better hydration of the protein surface, and salting in[12]. There has been considerable debate as to what precisely is meant by this statement. It feels unreasonable to suggest

[1]School of Biosciences, University of Sheffield, Sheffield S10 2TN, UK. [2]Present address: Certara UK Ltd, Level 2-Acero, 1 Concourse Way, Sheffield S1 3BJ, UK. ✉e-mail: m.williamson@sheffield.ac.uk

that kosmotropes could actually reduce the concentration of water in the protein/solvent interface: in that case, what exactly does this statement mean? We discuss this important question below.

Starting roughly in the 1960s, this theory was seen to have problems[10]. One of these is that under some circumstances (particularly at a pH lower than the pI of the protein) the Hofmeister effect is reversed (to produce the reverse or inverse Hofmeister effect) at low co-solute concentration: for example, low concentrations of sulfate increase protein solubility and decrease stability[13–15]. Several other mechanisms were proposed, of which the most popular are (1) Preferential hydration, which proposes (on the basis of detailed thermodynamic analysis, much of it initiated by Timasheff and further developed by Schellman[16]) that kosmotropes interact more strongly with water than with the protein, and are therefore preferentially excluded from the protein surface, leading to a preferential hydration of the protein surface in the presence of kosmotropes[17,18]. By contrast, chaotropes interact directly with the protein surface and so reduce protein hydration. This work was then extended into the concept of preferential interaction, suggesting that chaotropes accumulate at nonpolar (or less well hydrated[19]) protein surfaces[20–22]. Osmolytes are also proposed to be excluded from the protein surface[23]. Investigations have extended into more specific details of which cosolutes interact with which regions of the protein surface[24–28], with the extreme view being that Hofmeister effects relate to the direct interactions of cosolutes with the protein, and that water making/breaking is irrelevant[2]. (2) Volume excluded effect[16,29]. This theory is also based on a clear thermodynamic premise, which follows directly from Kirkwood-Buff theory (discussed below)[30], and notes that addition of a co-solute to a protein/water solution will inevitably reduce the volume available to the protein; and this in itself stabilizes the folded form of the protein, even in the absence of any interaction between protein and co-solute[31,32]. This cannot be the only effect of co-solutes, because protein stability and solubility clearly do not depend only on the volume of the co-solute.

The water maker/breaker explanation is a molecular mechanism, while the two more recent explanations are based on thermodynamics, irrespective of any molecular mechanism. They are therefore different types of explanation. Kirkwood-Buff theory relates interactions to thermodynamics and thus links the two, and we therefore discuss below how it may point towards a resolution of the problem.

One of the major problems in considering the molecular origin of co-solute effects is that it is difficult to measure binding directly (particularly when it is very weak) and to separate it from more general solvation effects. NMR chemical shifts are exquisitely sensitive to the local environment, and we show here that NMR can clearly distinguish binding (which produces a typical saturation curve) from solvent interactions (which are individually much weaker interactions and produce linear shift changes). The method is applied to the binding of barnase to a range of Hofmeister anions and osmolytes (osmolytes being well-solvated neutral molecules that are produced by cells to keep proteins functioning in stressful conditions, discussed in more detail below). The resulting data allow us to delineate clearly which effects are due to binding, and which are not. Barnase is a small RNA-binding protein. It therefore has a high pI, making it a good test model. We subsequently discuss how these results can be rationalized, concluding that essentially all the existing data, on both Hofmeister anions and stabilizing osmolytes, are consistent with a model in which Hofmeister ions bind weakly to the protein, changing the protein surface charge distribution and dynamics at low ion concentration (up to 100–200 mM, by which point the binding saturates), and thereafter only act to modulate solvent fluctuations, which has important thermodynamic consequences on the protein; whereas osmolytes interact very weakly with the protein, if at all, and act entirely by altering the solvent. This paper builds on our earlier work[33], which looked at the effects of single Hofmeister ions on barnase. The conclusions are similar. Here, the inclusion of results from osmolytes has widened the discussion and has allowed us to develop the idea of solvent fluctuation, rather than simple structural effects; it also demonstrates the importance of the binding of charged ions to the protein surface.

## Results

### An outline of the approach

Two-dimensional NMR spectra can be used to obtain information on different nuclei in a protein. $^{15}$N HSQC spectra provide a quick and convenient way to measure $^{15}$N and $^1$H$_N$ shifts, from backbone amides and the side-chains of Gln, Asn, Arg Nε, and Trp. Two-dimensional HNCO spectra give a sensitive and rapid measurement of carbonyl carbon shifts, while $^{13}$C HSQC spectra provide shifts from all C–H pairs. This combination of 2D spectra provides detailed information on protein structure and dynamics. We collected all of these spectra, acquiring one of each type of spectrum at each titration point as co-solutes were added. The measured shifts are available in summary form in Supplementary Data 1, and sample titrations for $^{13}$C′ and $^{13}$C-$^1$H are shown in Supplementary Figs. 1 and 2. By far the most informative data proved to come from the $^{15}$N HSQC spectra, and it is therefore these that we discuss in most detail below. This is partly because the shift changes in $^{15}$N and $^1$H$_N$ are large and easily measured, but mainly because the only chemical shift perturbation that can be related to a structural change with any confidence is that downfield changes in $^1$H$_N$ shifts arise very largely from an increase in hydrogen bond strength[34–36].

For each titration point, protein 2D peak positions were measured and loaded into a database. Typical titration data are shown in Fig. 1a. It is clear that the data do not fit a typical titration curve, which is a hyperbola with an asymptotic shift value. There is clearly a hyperbolic saturation curve, which reaches its asymptotic value at around 200 mM co-solute, but in addition there is added to this a second effect which appears linear out to at least 1 M. The relative magnitudes (and signs) of the hyperbolic and linear components vary widely. We hypothesized that the hyperbolic parts indicate direct binding of the co-solute to the protein, while the linear parts represent indirect effects on the protein chemical shifts caused by interactions of the co-solute with the solvent, which thus leads to altered protein solvation. Linear changes in shift could also arise from co-solute-induced conformational changes in the protein. Timasheff has pointed out that very weak protein/solute interactions (with a dissociation constant of the order of 1 M or weaker) are best considered as an exchange between water and solute, ie as a form of differential solvation[37]. Thus, very weak binding is thermodynamically equivalent to a change in solvation, implying that these two possibilities are two aspects of the same phenomenon. We therefore fitted the data to a sum of two equations: a hyperbolic curve plus a linear change. Fitting to the hyperbolic curve gives a resultant binding affinity, which can be tabulated as either a dissociation constant $K_d$ (with units of concentration) or an association constant $K_a$ (with units of reciprocal concentration). For most purposes, $K_d$ is the more natural value. However, it suffers from the disadvantage that larger $K_d$ means weaker binding, meaning that a histogram of $K_d$ vs residue number draws attention to the weaker binding, rather than stronger binding, which is normally of greater interest. Thus in the analysis below, we use $K_a$ when examining binding to the protein surface. We note that the linear change is in fact a binding curve with very weak affinity, and will therefore start to curve at some point. Based on the likely affinities, curvature is unlikely to become apparent until the co-solute reaches a concentration of several M.

For fitting shift changes to obtain the dissociation constant, one would normally use a quadratic equation[36]:

$$\Delta\delta_{\text{obs}} = \Delta\delta_{\text{max}}\{([P]_i + [L]_i + K_d) - [([P]_i + [L]_i + K_d)^2 - 4[P]_i[L]_i]^{1/2}\}/2[P]_I \tag{1}$$

However, when the binding is very weak, and in particular when the protein concentration $[P]_i$ is much less than both $K_d$ and the co-solute concentration, then this equation simplifies to the Langmuir isotherm:

$$\Delta\delta_{obs} = \frac{\Delta\delta_{\text{max}} \cdot [L]_i}{K_d + [L]_i} \tag{2}$$

where $\Delta\delta_{obs}$ is the observed shift change, $\Delta\delta_{\text{max}}$ is the maximum shift change on saturation, and $[L]_i$ is the ligand concentration at titration point $i$. This

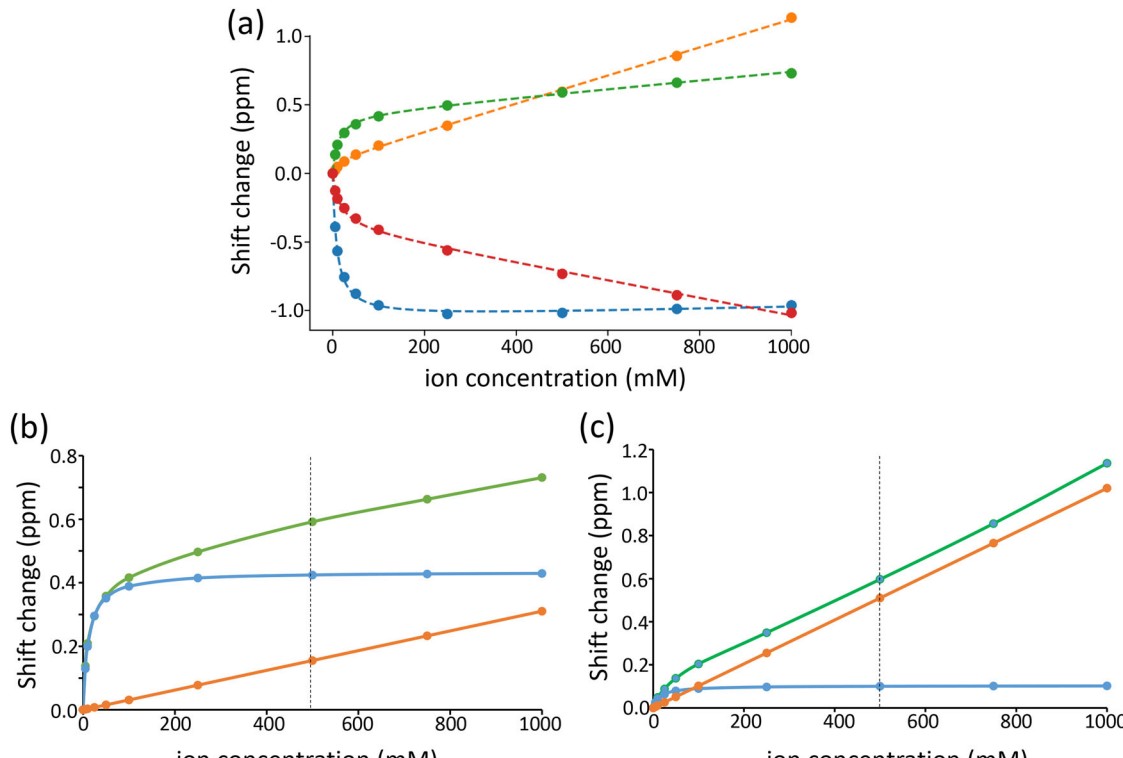

**Fig. 1 | Changes in $^{15}$N chemical shift for selected residues in barnase on titration with an equimolar mixture of SCN$^-$/Cl$^-$. a** Experimental shifts (points) and curves fitted to Eq. (3) (lines). (blue) S38 ($m_L$ 80 ppb/M, $\Delta\delta_{max}$ −1.062 ppm, $K_d$ 9.06 mM); (orange) W71 ($m_L$ 1020 ppb/M, $\Delta\delta_{max}$ 0.103 ppm, $K_d$ 15.46 mM); (green) G81 ($m_L$ 310 ppb/M, $\Delta\delta_{max}$ 0.434 ppm, $K_d$ 11.77 mM); (red) S85 ($m_L$ −628 ppb/M, $\Delta\delta_{max}$ −0.413 ppm, $K_d$ 15.56 mM). **b, c** Breakdown of shift changes observed in (**a**) for (**b**) G81 and (**c**) W71 into the linear component (orange), binding component (blue) and sum (green).

equation is simpler than (1) and gives more reliable fits. We have therefore used it throughout. The complete equation used here for fitting the experimental shift changes is therefore

$$\Delta\delta_{obs} = \frac{\Delta\delta_{max}.[L]_i}{K_d + [L]_i} + m_L[L]_i \qquad (3)$$

which gives three fitted values for each nucleus: $K_d$, $\Delta\delta_{max}$, and $m_L$. $m_L$ is the gradient of the linear component, and gives information on the indirect effect of the co-solute on protein chemical shifts via its effect on solvent. The same equation has been used by others, mainly Cremer and collaborators[19,28,38]. We discuss later the significance of $m_L$: for present purposes, it is an empirical parameter that describes how the co-solute affects the protein indirectly via solvent. We devoted considerable effort to making sure that the values fitted were reliable: for example, that fitted $K_d$ values had genuine meaning as affinities. Further details can be found in the Methods section.

The four examples shown in Fig. 1a demonstrate that a simple measurement of a shift change at some fixed co-solute concentration is not very informative. To illustrate this point further, in Fig. 1b, c we show two datasets for $^{15}$N chemical shift changes with the addition of SCN$^-$/Cl$^-$. At 500 mM co-solute (dashed line), the measured chemical shift changes are almost identical at 0.6 ppm. However, it is clear from the curves that the origins of these shift changes are very different: the two residues bind with similar affinity, but the $\Delta\delta_{max}$ for G81 is much larger, and the $m_L$ value is much smaller. At lower co-solute concentrations the shift changes are more closely related to $\Delta\delta_{max}$, but this is the least useful of the three fitted parameters, because chemical shift changes are very difficult to ascribe directly to a structural change[36]. Almost the only generalization that one can make about chemical shift changes is to say that a downfield shift of an amide proton indicates an increase in the strength of hydrogen bonding involving that proton[35]. We therefore suggest that any measurement of co-solute

binding needs to be done using titration data and fitting to a curve such as Eg. (3), rather than measuring a shift change at a single co-solute concentration.

In this work, we use chemical shift changes to provide information on binding and solvent effects, using co-solute concentrations ranging from zero to 1 M. In order to avoid possible confounding factors, we tried to limit the number of species present in the solution, which ideally should consist of just the protein and the co-solute(s). In particular, we tried conducting the titrations with no buffer present, effectively using the protein as its own pH buffer. However, this produced two problems. First, the pH meter gave very erratic and unreliable readings without buffer present. Second, three signals in the $^{15}$N HSQC (L42, R83 and the side-chain signal of W35) had enormously variable chemical shifts, even in solutions with nominally the same pH, and all prepared and dialyzed extensively in the same HPLC grade water. These three amides are adjacent in the structure (at the edge of the substrate binding pocket), and in the crystal structure take part in a hydrogen bonding network involving two water molecules (Fig S3). We therefore speculate that this hydrogen bond network may be very sensitive to solution conditions, and we do not consider the results for L42 and R83 further. For these reasons, all experiments were carried out using 5 mM acetate, pH 5.8, conditions that gave reproducible peak positions while using only a very low concentration of additional solutes.

## Titrations with the Hofmeister anions thiocyanate, chloride, and sulfate

Titrations were conducted on barnase using sodium thiocyanate, sodium chloride, and sodium sulfate. The sodium ion was used in each case because it has been shown to have very little effect on proteins, and is in the middle of the cation Hofmeister series[10]. Chemical shifts of $^1$H$_N$, $^{15}$N, and $^{13}$C' were measured and fitted to Eq. (3). Details are provided in Methods. We analyze the results, presenting first the binding affinities (presented as $K_a$ for visual

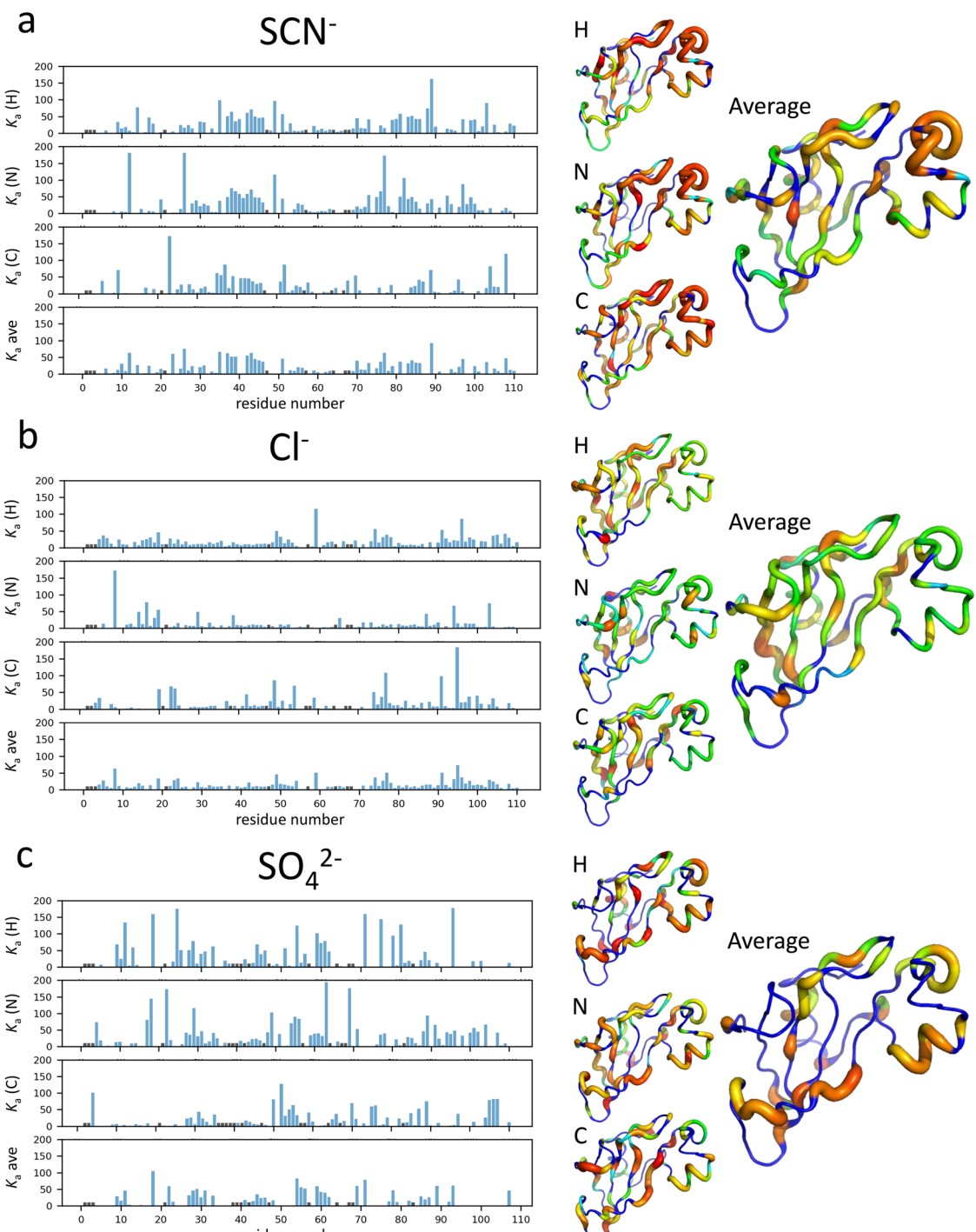

**Fig. 2 | Apparent association constants $K_a$ for the binding of Hofmeister ions to individual amides in barnase.** Histograms are (**a**) thiocyanate, (**b**) chloride, (**c**) sulfate, and show the fitted $K_a$ for binding to amide proton, amide nitrogen and backbone carbonyl (from the previous residue), together with the mean. Residues for which no $K_a$ could be fitted are shown with black bars. $K_a$ values are indicated on the structures on the right, first as individual H/N/C and then as the mean. Residues for which there is no data are shown as blue, and increasingly strong affinities are shown as redder colors and thicker tubes. Residue numbering can be found in Fig. S4.

clarity) and then the $m_L$. As noted above, the most informative shifts were those of $^1H_N$.

Fitted values of $K_a$ are inherently noisy. We therefore averaged over $^1H_N$, $^{15}N$, and $^{13}C'$ affinities (with the carbonyl values included within the same peptide bond, ie with the H and N of the following residue, because chemical shift changes tend to be correlated within the same peptide bond[39]). The results are shown in Fig. 2.

It is clear from the data in Fig. 2a that there is an extended binding site for thiocyanate, comprising the residues in the loop at top right (residues

35–46), and several adjoining regions across the top of the structure, in particular residues 81–86, which have consistent patterns of strong affinity across all nuclei. The average value of $K_a$ for these residues is about 45 M$^{-1}$ (equivalent to a $K_d$ of 22 mM), which is a reasonable value for a charged ion binding to a protein surface, and is inside the range of literature values[14,38,40].

We note that the data in Fig. 2 are all for nuclei in the peptide backbone. Measurements were also made using $^{13}C$ HSQC, which gave shift changes for aliphatic H and C. Shift changes were small and almost entirely linear (except for Hα and Cα) [Supplementary Data 1, see Methods for linearity

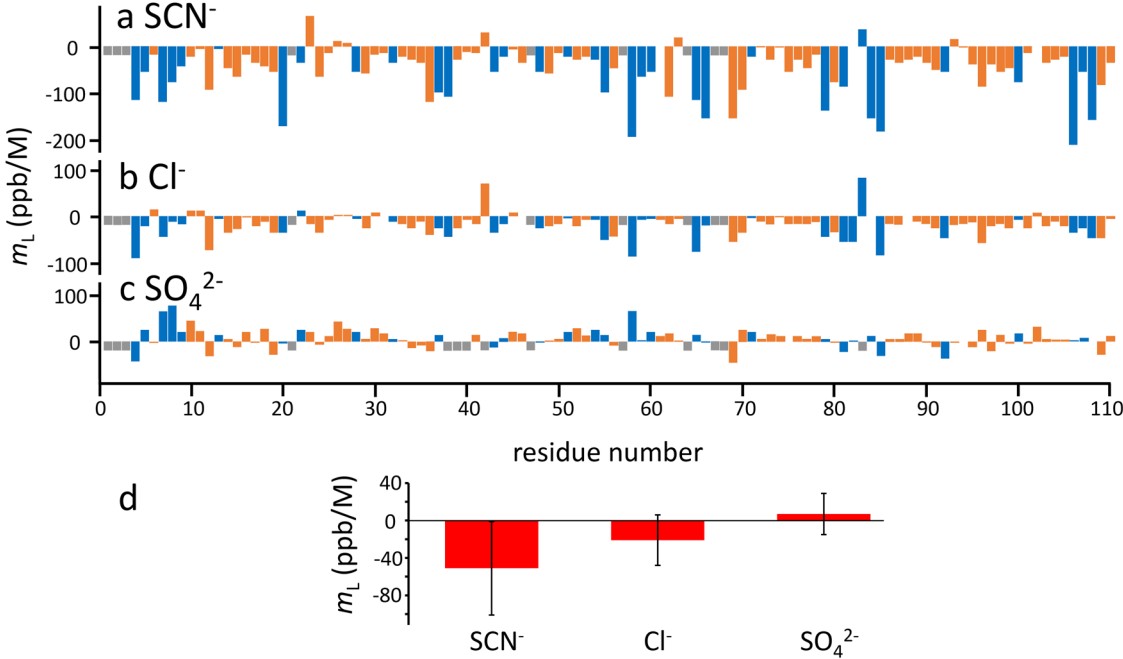

**Fig. 3 | $m_L$ for amide protons in barnase on titration with single Hofmeister ions. a–c** values for thiocyanate, chloride and sulfate respectively. Buried amides are shown in orange, and surface-exposed amides in blue. Residues for which no $m_L$ was fitted are in gray. **d** Distribution of $m_L$ for each ion, showing means and standard deviations ($n = 101$, 101, and 97 for SCN⁻, Cl⁻, and SO₄²⁻ respectively).

criteria], strongly implying that affinities of Hofmeister ions for aliphatic groups were too weak to measure, mitigating against the theory that chaotropes exert their effects by binding to hydrophobic patches.

The affinities were fitted assuming a single binding site with dissociation constant $K_d$. If there are $n$ equivalent binding sites, then the equation is essentially identical[41], meaning that it is not possible to distinguish by NMR whether there is one binding site or many. However, a dissociation constant of about 22 mM seems entirely reasonable for a single ion. Our fitting indicates binding to a region of the protein surface rather than to a single site, described as "territorial binding"[42] as opposed to site-specific binding. Territorial binding represents weak associations of ions with regions of high charge density, in which the ion diffuses rapidly on the surface before finally dissociating[43–45]. This region of the protein is not the most positively charged patch (which is the active site of this RNAse), but it is close to the active site and to other positively charged regions. Thus, our hypothesis is that a single thiocyanate ion binds to a positively charged patch on the protein surface, with an average dissociation constant of about 22 mM. This behavior is consistent with molecular dynamics calculations of ions binding to proteins[46].

By contrast, for chloride, there is no clearly defined binding location, and overall weaker binding (Fig. 2b). This is not very surprising, given that it is a single spherical and well-hydrated ion.

Sulfate binding (Fig. 2c) is similar to thiocyanate. There is a clearly defined binding region at residues 27–31 and 54–62, which are adjacent in the structure, and form a strip across the bottom of the structure as shown in Fig. 2c, with an average affinity also of 22 mM. We therefore conclude that sulfate binds in a similar way to thiocyanate: a single ion binds, moves around on this positive surface, and eventually dissociates. The location is different from that of thiocyanate, presumably because the two ions have different shapes and preferred modes of binding to the different groups on the surface.

We also fitted the values of $m_L$ and analyzed them. We fitted $m_L$ for ¹H$_N$, ¹⁵N, and ¹³C'. The values for ¹⁵N and ¹³C were not easy to interpret. We attribute this to the observation that ¹H$_N$ shift changes have a fairly straightforward origin[36]: stronger hydrogen bonding tends to produce a downfield shift. By contrast, N and C shifts are more complicated. First,

because the size and direction of the shift change depend on the position of the hydrogen bonding group relative to the amide bond; and second, because hydrogen bonding to a carbonyl group causes large shift changes to the ¹⁵N within the same amide bond, and vice versa, implying that one cannot easily identify which face of the peptide bond is interacting with co-solute based on the shift change of N and C, whereas amide proton shifts are largely affected only by hydrogen bonding to the proton. We therefore concentrate here on amide protons: effects on other nuclei are shown in Supplementary Material.

The results are shown in Fig. 3. There is a wide range in individual $m_L$, but the three anions have clearly different ranges of $m_L$: thiocyanate $-51 \pm 51$ ppb/M, chloride $-16 \pm 26$ ppb/M, and sulfate $+7 \pm 22$ ppb/M, which are significantly different distributions ($p < 0.001$, Student's two-sided $t$ test). We note that these values are in the same order as the Hofmeister series: sulfate at one end and thiocyanate at the other, with chloride in the middle; and that the amide nitrogen gradients have the same behavior. We therefore propose that the ion-specific Hofmeister effect is a result of the indirect solvent-mediated effect (measured by $m_L$), while the binding events are a distinctly different phenomenon. Further support for this view comes from the fact that Hofmeister effects are widely held to be directly proportional to co-solute concentration, up to and beyond 1 M[2,47]. This is clearly true for the solvent-mediated effects, but not for the binding, which have $K_d$ values in the range 20–50 mM, and are therefore saturated by 100–200 mM. Chemical shift changes to aliphatic ¹³C-¹H signals (Supplementary Data 1 and Fig. 3) show that there is no greater binding of chaotropes to hydrophobic regions than of kosmotropes (and in fact no clear evidence for chaotropes binding to hydrophobic regions at all). For these reasons, we propose that the preferential hydration/interaction theory is not a helpful explanation for the Hofmeister effect. We return to these considerations later.

Figure 3a–c also presents a breakdown of $m_L$ by whether the amide is exposed or buried. Distributions for the two groups are essentially identical, for all three ions. This was a surprise. A positive ¹H$_N$ $m_L$ for exposed amides (as found for the kosmotrope sulfate) implies that titration with sulfate increases the strength of hydrogen bonding from the surface of the protein to water. There is a good body of evidence[48,49] that shows that kosmotropes do increase the ordering of water around the kosmotrope ion, and

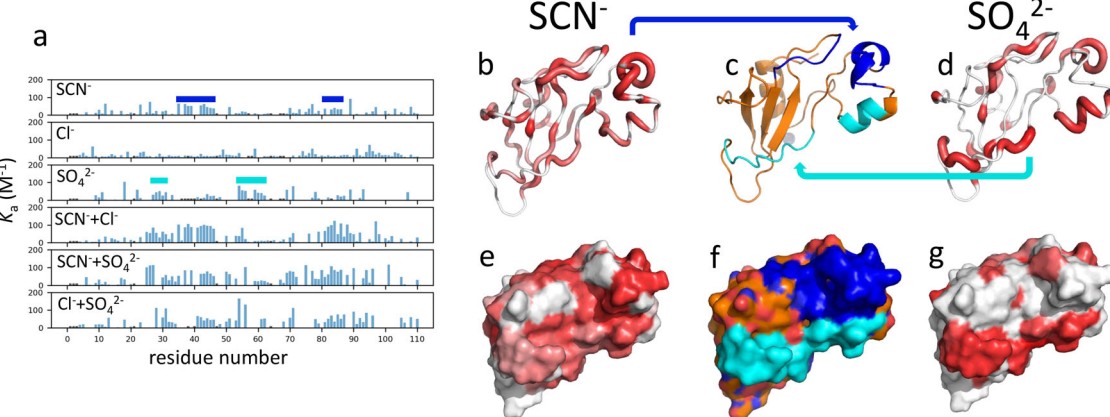

**Fig. 4 | Affinity data for pairs of Hofmeister ions. a** Apparent association constants $K_a$ for the binding of Hofmeister ions to individual amides in barnase, with the Hofmeister ions indicated. The top three graphs show the data for the three individual Hofmeister ions, reproduced from Fig. 2, and collected together to show the $K_a$ averaged over H, N, and C' shifts. Dark blue bars indicate the binding sites for thiocyanate, and cyan bars indicate the binding sites for sulfate. The bottom three graphs show the corresponding $K_a$ averaged for the three pairs of ions. On the right are shown the individual binding sites for (**b**) thiocyanate and (**d**) sulfate, with (**c**) showing binding for the pair of ions. **b** sites previously identified as thiocyanate binding sites are colored in shades of red with increasingly thick tubes (compare Fig. 2a), which are marked on the protein surface in (**e**). These indicate a thiocyanate binding region at the top right of the protein, which is indicated in dark blue in (**c**) and (**f**). Similarly, (**d**) and (**g**) show the sulfate binding sites which form a strip across the bottom of the protein, indicated in cyan in (**c**) and (**f**) (compare Fig. 2c).

chaotropes decrease it[50], and that they also have a long-range effect on the solvent; for example a change in the radial distribution function of the O-O distance in bulk water[51], such that bulk water is more ordered in the presence of kosmotropes. The standard Hofmeister kosmotrope/chaotrope theory says that by ordering the water, kosmotropes act to withdraw water from the protein surface, and that it is this behavior that makes proteins more stable but less soluble[24]. On this basis, we would expect a kosmotrope to produce a negative $m_L$ for the surface-accessible amides, which is the exact opposite of what we observed. Proteins are large and complicated dynamically coupled assemblies, and have the property that weakening of noncovalent interactions in one region (for example, the protein/solvent interface) is compensated for by increasing the strength of interactions elsewhere (for example, inside the protein)[52]. This would lead us to expect that buried amides would have $m_L$ with opposite sign to the $m_L$ of surface-exposed amides, and thus for example that the orange bars in Fig. 3 would tend to go in the opposite direction to the blue bars, which is clearly not the case. This means that the standard kosmotrope/chaotrope theory has two problems: it predicts $m_L$ for surface amides that have the wrong sign; and it also predicts that surface and buried amides should have oppositely signed $m_L$, when clearly they do not. We are therefore forced to conclude that the conventional explanation of kosmotropes withdrawing water cannot be correct, and that we need a fundamentally different approach in order to explain these observations. This different approach is discussed below in the Discussion.

### Titrations with pairs of Hofmeister ions

One way to test the validity of our model is to combine Hofmeister ions in pairs. Thus for example if the $m_L$ are a measure of the effect of ions on water structure, then one might expect the $m_L$ for pairs of ions to be simple sums of the corresponding individual $m_L$, on the basis that the concentration of water (56 M) is so much greater than the concentrations of the ions that there is unlikely to be significant ion-ion interaction at concentrations up to 1 M. By contrast, binding affinities are unlikely to behave in such a simple additive way.

The results for additivity of binding affinities are shown in Fig. 4, where we show the effects of combining the three Hofmeister ions in pairs. In these experiments, ion pairs were added as equimolar mixtures, so that for example 1 M [sulfate/thiocyanate] means 1 M sulfate and 1 M thiocyanate. If each ion binds independently, with no interactions between them and no change in protein structure or dynamics, then the affinities for each ion would be expected to be unchanged from the values measured for the

individual ions. We focus on the thiocyanate/sulfate pair because there are clear binding sites for these ions taken individually. These results show that, as expected, titration with sulfate and thiocyanate together gives rise to chemical shift changes and measurable affinities at both binding sites. The same is approximately true for the other pairs: any titration involving thiocyanate produces strong affinities at typical thiocyanate sites, and titrations with sulfate give at least some of the sulfate-specific binding interactions. In other words, the addition of a mixture of sulfate and thiocyanate produces binding of sulfate at preferred sulfate-binding sites, and of thiocyanate at preferred thiocyanate-binding sites, with no obvious interference between the ions. There is a second conclusion that can be made from these results, namely that titration with a pair of ions gives rise to a stronger affinity than each ion alone. This observation is discussed below.

Analysis of the $m_L$ is shown in Fig. 5. As with single ions, there is a large variation in individual $m_L$ (Fig. 5a–c), but the distributions are characteristic for the different pairs (Fig. 5d), and (within error) match the sums of the individual values. Thus for example, the average $m_L$ for the SCN⁻ + Cl⁻ pair is approximately the sum of the $m_L$ for SCN⁻ and Cl⁻ individually (Fig. 5d), as would be expected if the $m_L$ report on bulk solvent effects. This is more clearly seen in Fig. 5e–g, where for each pair of ions, there is remarkably good correlation between the residue-specific sum of the $m_L$ for the two ions separately and the $m_L$ for the mixture of both ions together.

We conclude that the results for pairs of ions exactly match our hypothesis. The locations of binding sites are largely retained, and there is some positive cooperativity of anion binding, in that the binding of two ions always produces affinities stronger than the two ions separately. The $m_L$ simply add independently, as would be expected for a solvent-mediated interaction. These results reinforce our confidence in our interpretation of the binding curves, that the hyperbolic curves arise from direct binding interactions, while the linear parts are indicative of indirect solvent-mediated interactions, roughly as expected for a chaotrope/kosmotrope theory.

### Titrations with osmolytes

Osmolytes form an interesting contrast to Hofmeister ions. Chemically, they are very diverse: they include methylamines such as trimethylamine N-oxide (TMAO), glycine betaine, and sarcosine; amino acids and their derivatives such as proline and ectoine; polyols and sugars; and urea[53]. They are mostly overall neutral, and are small and soluble. They are produced by organisms that live in stressful situations, such as high salt, high pressure, or extremes of pH or temperature, and their function is to keep the cells

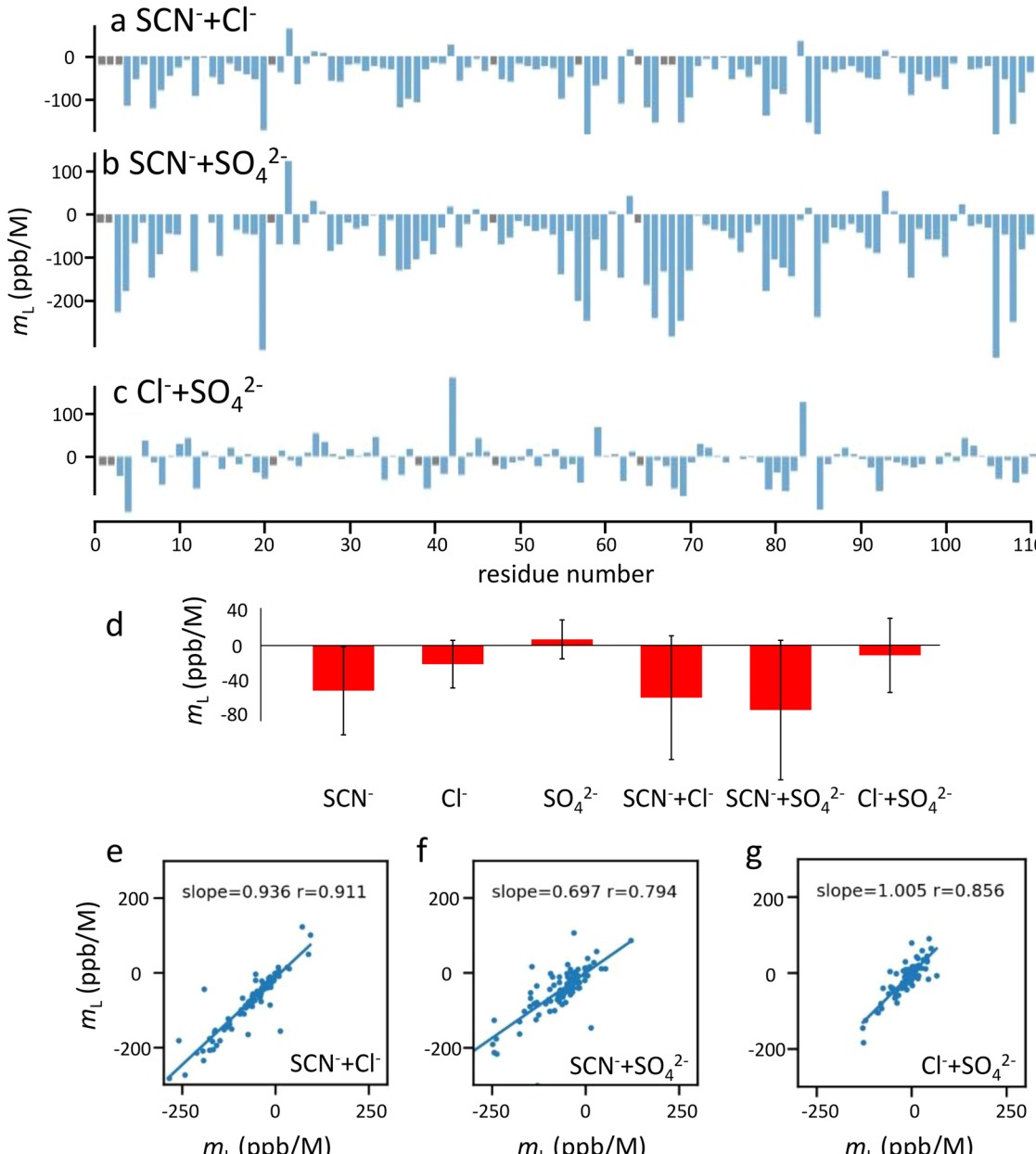

**Fig. 5 | $m_L$ for amide protons on titration with pairs of Hofmeister ions.**
**a–c** Values for ion pairs. Residues for which no $m_L$ was fitted are in gray.
**d** Distribution of $m_L$. Values for the individual ions are copied from Fig. 3. The mean values and standard deviations for the pairs are $-59 \pm 70$ ppb/M for SCN$^-$/Cl$^-$, $-73 \pm 79$ ppb/M for SCN$-$/SO$_4^{2-}$, and $-11 \pm 42$ ppb/M for Cl$-$/SO$_4^{2-}$ ($n = 101$, 105, 103 for these three respectively). **e–g** Correlation of $m_L$ by residue: comparison of (**e**) sum of $m_L$ for SCN$^-$ plus $m_L$ for Cl$^-$ (y-axis) compared to $m_L$ for mixture of SCN$^-$ and Cl$^-$ (x-axis) (**f**) same comparison for SCN$^-$ and SO$_4^{2-}$; (**g**) same comparison for Cl$^-$ and SO$_4^{2-}$. The gradients of the lines of best fit and Pearson correlation coefficients of the comparisons are indicated on each plot ($n = 101$, 105, 103 for (**e**), (**f**), and (**g**), respectively).

working: essentially to maintain protein function as close to "normal" as possible. They therefore act to "stabilize" proteins in a broad sense[54]. Here, we have studied three osmolytes (Fig. 6). TMAO (Fig. 6a) is overall neutral (with one negative and one positive charge), and is produced by many marine organisms, both to counteract high osmotic strength and to cope with the high pressures lower down in the ocean[55]. Ectoine (Fig. 6b) is produced by halophilic bacteria and used to counteract high osmotic strength, where it is also referred to as a compatible solute[56–58]. Betaine (N-trimethyl glycine; Fig. 6c) is again neutral with one positive and one negative charge, and is accumulated to counteract osmotic stress and high temperature. It also improves the thermal stability of proteins[59]. There are many reports that osmolytes are excluded from protein surfaces, and a number of suggestions that this is the basis for their function[60,61].

In the same way as was done above for Hofmeister ions, we titrated osmolytes into barnase, and acquired a range of 2D spectra to follow chemical shift changes for H, N, and C nuclei. As commented above, the most useful of these nuclei proved to be amide protons, so although we have data for other nuclei (summarized in Supplementary Material) we report here only on amide proton shifts.

The most striking result from the osmolytes was that most residues showed no evidence of binding at all, as compared to Hofmeister anions where the majority of residues had significant curvature in their chemical shift profiles that could be fitted to a $K_d$. These results are shown in Fig. 7. TMAO has a reasonable number of residues that have measurable binding, but ectoine has only two (these being L42 and R83, described in Figure S3 as being peculiar in their behavior), and betaine has none at all: in other words,

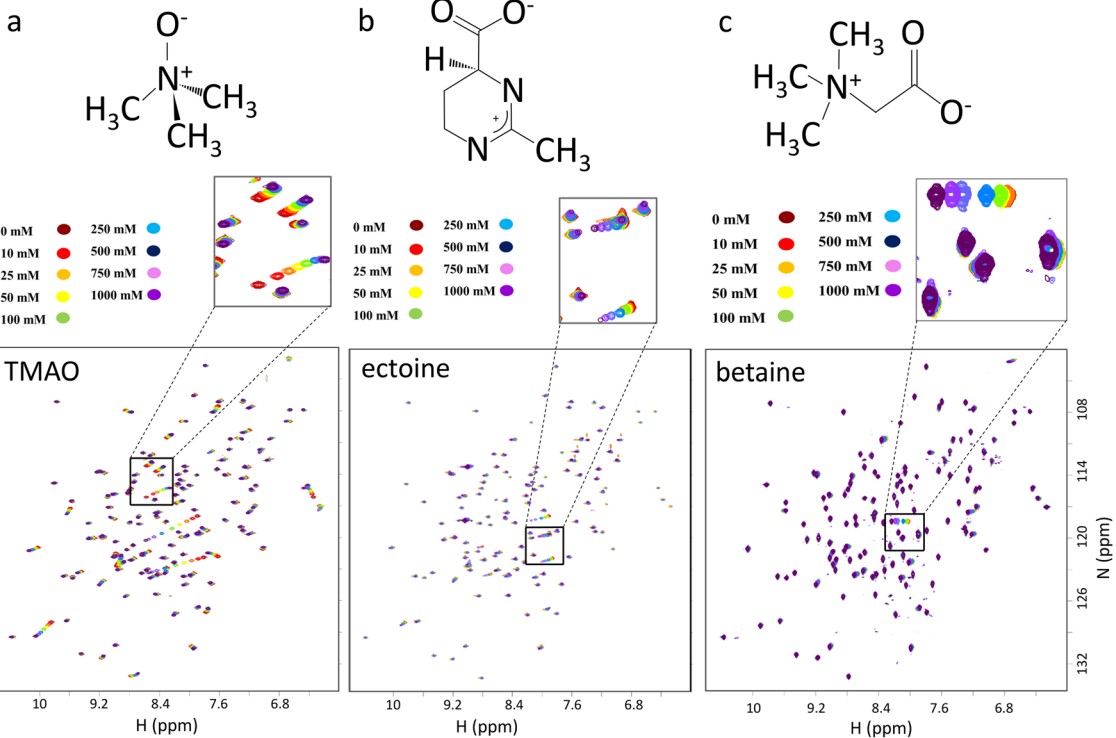

**Fig. 6 | HSQC titrations of barnase with osmolytes.** (**a**) TMAO, (**b**) ectoine and (**c**) betaine.

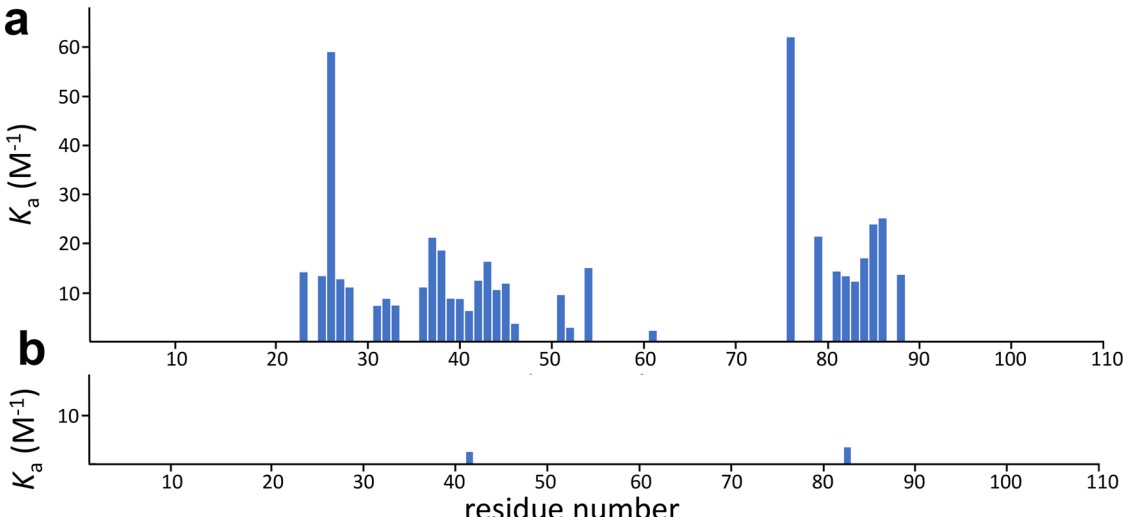

**Fig. 7 | $K_a$ values for titrations of barnase with osmolytes.** (**a**) TMAO, (**b**) ectoine. No data are shown for betaine because there were no residues that could be fitted with confidence to a binding curve.

the NMR spectra suggest that there is no binding of betaine to barnase, very little binding of ectoine, and only very weak binding of TMAO. The same is also the case for $^{15}N$ and $^{13}C'$ nuclei, which also show no evidence of binding to betaine. The average $K_a$ for TMAO over all nuclei is approximately 60 mM; for ectoine there are few residues to fit, which give an average value of about 300 mM; and for betaine there is no evidence of any binding. Thus, the binding of all the osmolytes is weaker than for the Hofmeister ions, and it affects far fewer residues.

However, almost all residues had chemical shift changes that were linear with concentration and could be fitted to give $m_L$, for all three osmolytes. The results are shown in Fig. 8 (summarized in the form of histograms), which also compares the distributions to those seen for Hofmeister ions. As seen for Hofmeister ions, there is a reasonably wide spread

of individual values, but a fairly well-defined distribution, allowing us to characterize the average $m_L$ for TMAO as $+9.7 \pm 16.9$, ectoine $+16.5 \pm 21.3$, and betaine $+4.3 \pm 15.9$ ppb/M. The most obvious conclusion to be drawn from these results is that the osmolytes have $m_L$ very similar to each other and to those of sulfate: in other words, from a comparison of solvent-mediated effects (which we argue above are the effects mainly responsible for Hofmeister effects), the osmolytes behave almost identically to sulfate, and should therefore be considered to be kosmotropes.

Figure 8gh compares the distribution of $m_L$ averaged over all three osmolytes, for buried and exposed residues. There is a wider spread of values for the exposed residues, but the mean is around $+10$ for both distributions, confirming the observation made above that buried and exposed residues have very similar $m_L$ distributions.

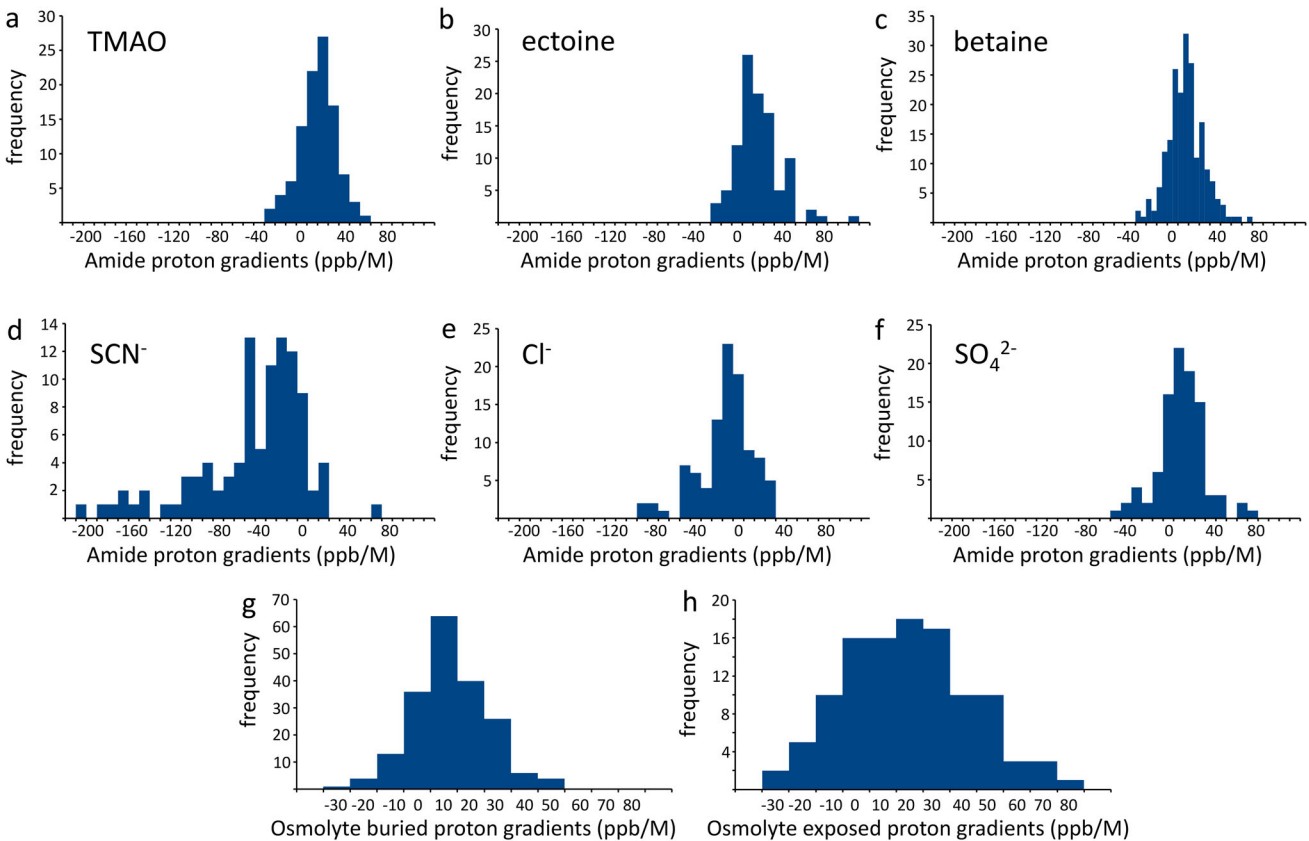

**Fig. 8 | Distribution of $m_L$ for addition of co-solutes to barnase. a–c** Osmolytes, **d–f** Hofmeister ions. **g, h** compares the distribution for all three osmolytes taken together, for buried and surface-accessible amides.

The $m_L$ arise from interactions between co-solutes and water, which then goes on to affect solvation at the protein surface. One would therefore expect that the $m_L$ for each amino acid will correlate for different co-solutes, because they arise from the same physical origin. A selection of such correlations is shown in Fig. 9.

All of the sets of $m_L$ display a convincing correlation, with that between ectoine and betaine being particularly good (Fig. 9a, $r = 0.93$). We interpret this result as confirming that the $m_L$ values measure an effect mediated by a common mechanism, namely the effect of co-solutes on the water. Not surprisingly, the correlation is strongest when the co-solutes do not bind to the protein (ectoine and betaine).

Thus, in summary, the osmolytes give $m_L$ similar to those of sulfate, and would therefore appear to be acting as kosmotropes. The big difference compared to Hofmeister ions is that they bind weakly to the protein, with betaine showing no evidence for any direct interaction. The effects of osmolytes on protein stability are therefore due entirely to solvent-mediated interactions, whereas Hofmeister ions have effects via the solvent and bind to the protein, changing its charge and its dynamics.

## Discussion

We have presented extensive sets of data that allow us to separately characterize the binding and solvent effect of three Hofmeister ions and three osmolytes on barnase. These data allow us to reach some conclusions on the effects of co-solutes, and to propose some hypotheses.

All three Hofmeister ions bind with low affinity; sulfate and thiocyanate have a dissociation constant of about 22 mM, with chloride weaker at about 70 mM. This means that they are fully bound by about 200 mM. Sulfate and thiocyanate bind mainly at a defined patch (with the two ions binding in different and non-overlapping locations), which is positively charged although not the most highly positively charged region on the surface. Chloride has no obvious binding patch. We have suggested that the

binding is likely to be one single ion at a time, which diffuses within the patch before dissociating. There have been a number of studies of ions binding to protein surfaces: see for example Yu et al.[45] and da Rocha et al.[62], which use molecular dynamics, Poisson–Boltzmann calculations and experiment, and reach similar conclusions from all three methods. They find that for a monovalent ion, the total ion excess (ie the difference between the number of ions close to the protein surface and the number of ions in an equivalent volume away from the protein surface) is roughly $z/2$, where $z$ is the total charge on the protein. The total charge of barnase at pH 5.8 is approximately +3.6 as calculated by Prot pi, so on this basis one would expect roughly 2 negatively charged ions close to the protein surface at any one time. Our finding of one thiocyanate or one sulfate close enough to cause appreciable chemical shift changes is therefore consistent with both theoretical and experimental expectations.

Titrations with pairs of ions show that binding of pairs of ions is synergistic in that the presence of one ion strengthens the binding of the other. These ions are negatively charged, and our titrations were conducted at pH 5.8, which is well below the pI of barnase (9.2), meaning that barnase is positively charged at pH 5.8. Thus, binding of one negatively charged ion reduces the overall positive charge on the protein surface. The binding patches for sulfate and thiocyanate are both on the same face of the protein, which also contains the highly positively charged active site. The binding of an anion to either of these patches will therefore overall reduce the electric dipole of the protein. Thus, the synergistic effect cannot be due to the change in surface charge or electric dipole, both of which would weaken the binding of a second anion. We propose that the synergy is instead an entropic effect: the binding of an ion, even when that binding is spread over a large surface patch, reduces fluctuations at the surface[63], and thus overall reduces the fluctuations across the entire protein. In turn, this makes binding of a second ion easier, because binding to a better defined surface requires less compensation between entropy and enthalpy. This is a good example of a

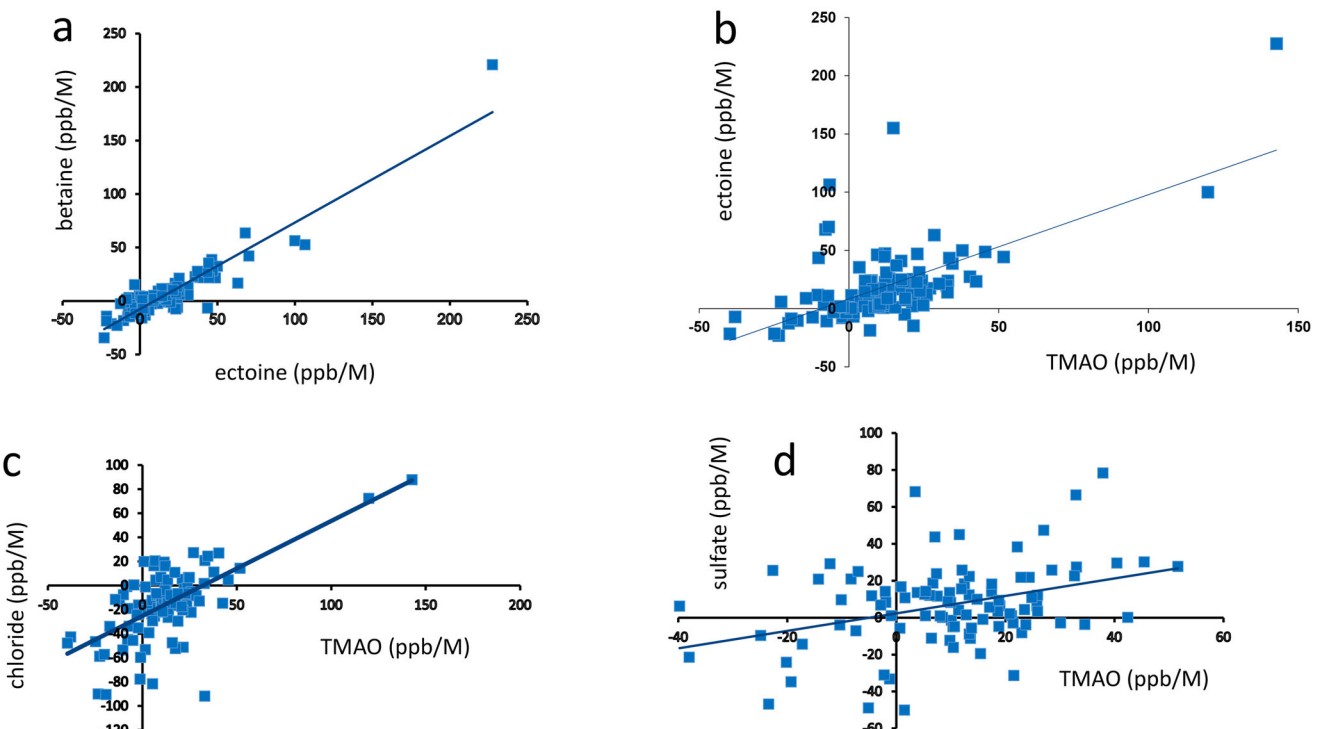

**Fig. 9 | Correlation of $m_L$ for individual amide protons with different co-solutes. a** Betaine vs ectoine, **b** ectoine vs TMAO, **c** chloride vs TMAO, **d** sulfate vs TMAO. Lines of best fit are indicated.

fluctuation affecting thermodynamics—an effect that is discussed in greater detail below.

We note that there has recently been an alternative explanation proposed for non-additive ion effects, specifically for effects on the lower critical solution temperature of poly(*N*-isopropylacrylamide)[64]. This work proposes that a well-hydrated anion such as sulfate withdraws counterions from a poorly hydrated anion (eg iodide or thiocyanate) and thereby increases its solubility; but at high concentrations of the poorly hydrated anion, sulfate forces it out of solution to the polymer surface. This effect may certainly contribute to the synergistic anion binding affinities observed here. However, we feel it is unlikely to be a major effect, because it was observed for all combinations of anions, including the chloride/sulfate pair, for which one could not describe chloride as poorly solvated.

The $m_L$ describe an effect of co-solutes on barnase that is linear with concentration up to at least 1 M; it is additive (a mixture of two co-solutes has an effect that is the sum of the solutes separately); it is highly correlated for pairs of ions (Fig. 9), indicating a common mechanism; and it goes in the Hofmeister order of thiocyanate—chloride-sulfate All of these features support the argument that the $m_L$ are closely related to the Hofmeister effect, and therefore imply that the Hofmeister effect arises from the indirect effect that solutes have on the water. However, this does not mean that the binding is irrelevant, as discussed below. We note that all three osmolytes have a $m_L$ similar to that of sulfate, suggesting that they all act as kosmotropes.

We noted above that the chemical shift changes seen for amide protons in barnase are not those expected based on the standard kosmotrope/chaotrope theory as described in the Introduction. According to this theory, a kosmotrope should order water, and should therefore reduce the average strength of hydrogen bonding from water to protein, because the water is "withdrawn" from the protein surface. It should therefore cause upfield shifts to amide protons. For the same reason, if hydrogen bonding to the surface is reduced, then hydrogen bonding in the interior should be stronger, so shift changes should go in the opposite direction to their direction for surface exposed amides. In fact, the observation is that buried and surface accessible amide shifts change in the same direction, which for

the kosmotropes is to go downfield (ie stronger hydrogen bonding). How can this be explained?

A coherent and realistic explanation, which also lends itself to further investigation, is provided by statistical thermodynamics, particularly as realized in Kirkwood-Buff (KB) theory[30,31]. This theory relates the interactions between solvent, solutes and co-solutes to chemical potential and thus to free energy, and thus potentially provides a mechanistic interpretation of the thermodynamic properties[30,65]. It is not a simple theory, and we particularly recommend Abbott[32] (available online as a free eBook) as a simple and readable introduction. We note two consequences of KB theory. (1) KB theory emphasizes Kirkwood-Buff integrals, $G_{ij}$, which are the integrals of the radial distribution functions between molecules i and j, where i and j could be water, protein, or solute. Of these, the most important for protein solubility is usually that between protein and co-solute[31,32]. Significantly, this does <u>not</u> mean that protein and co-solute need to interact, because a large negative $G_{ij}$ is created simply by a volume-excluded effect. This means that any co-solute added to a protein solution will inevitably increase the stability of the protein merely by taking up space, provided that the co-solute does not interact with the protein (for example urea, for which the binding interaction overcomes the volume exclusion effect). (2) A less commonly discussed consequence of KB theory is that the free energy of the different components in the solution is determined not just by the pairwise interactions between the different components present (which is more or less a structural effect), but also and equivalently[66] by the fluctuations of these interactions, specifically that large fluctuations give rise to correspondingly large Kirkwood-Buff integrals and thus large changes in free energy, which has been described as an "entropic force"[67]. On this view, kosmotropes order water molecules, and specifically they suppress the fluctuations of water. This behavior alters the chemical potentials of solutes, and leads to salting out of co-solutes (for example, protein). Chaotropes have the opposite effect: they enhance solvent fluctuations, which leads to a salting in of solutes. The important insight produced by this approach is to emphasize fluctuations rather than structural aspects. Thus, kosmotropes reduce the solubility of proteins not so much because they change the solvent structure and "withdraw" waters from the protein surface[31], but because they reduce

solvent fluctuations[13,68]. Lower solvent fluctuations reduce the ability of the solvent to hydrate the protein surface, and therefore make the protein less soluble. In this context, it is worth repeating a point made by Schellman many years ago, that very weak binding can just as well be regarded as preferential solvation by the solute[30]: when binding of co-solutes to protein is very weak, it makes more sense to consider it as a change in solvation.

This viewpoint immediately makes sense of the chemical shift changes. Kosmotropes reduce the solvent fluctuations: one effect of this, via the slaving of protein fluctuations to solvent fluctuation[69], is to reduce the solvent-induced protein fluctuations, in other words to transfer the internal energy of the protein more from entropy to enthalpy. An immediate consequence of this is that all hydrogen bonding interactions, both internal and with solvent, become more enthalpic—the hydrogen bonds "get stronger". This explanation thus immediately makes sense of the puzzling directions of chemical shift changes seen both with Hofmeister ions and with osmolytes.

We note that this explanation also suggests an answer to a key question, namely what is the molecular basis for the Hofmeister effect? The insight provided by Kirkwood-Buff theory is that ions on the left of the series are indeed kosmotropes, and ions on the right are chaotropes. However, we need to understand a kosmotrope not as something that withdraws water from the protein but as a co-solute that reduces solvent fluctuations, and a chaotrope as one that increases fluctuations. The language of removing water is not helpful. By seeing the action of kosmotropes as a global reduction of solvent fluctuations, we can also appreciate better that we should not seek an explanation at the level of individual interactions or hydrogen bonds: this is an effect based on ensemble fluctuations rather than structures. It is clear that this hypothesis is open to testing by a wide range of experimental and computational means: and in support of our proposals, many groups have observed for example reductions in the dynamic fluctuations of both water and protein in the presence of osmolytes/kosmotropes[70–77]. An obvious experimental test in this system is to measure the [15]N NMR order parameter of barnase in the absence and presence of osmolytes, which we are currently doing. We emphasize that understanding the activity of kosmotropes/chaotropes as a manifestation of fluctuations does not mean that it is wrong to use the language of structure or thermodynamics: merely that fluctuations simplify and unify the explanation, in roughly the same way that the Copernican viewpoint of heliocentric planetary motion is a simpler explanation than a geocentric one.

A helpful aspect of this proposal is that Kirkwood-Buff theory relates thermodynamic phenomena (such as chemical potential – the free energy of individual components in solution) to physically measurable parameters: in particular the Kirkwood-Buff integral, which is the spatial integral over the pair correlation functions, and can be determined by scattering, compressibility, or molecular simulations, thus allowing a direct testing of these proposals.

All three osmolytes have $m_L$ that are very similar to those of sulfate—in other words they behave like kosmotropes in their effect on water. Similar observations have been made before: for example, the radial distribution function of water in the presence of TMAO indicates a greater ordering of water[51]. However, sulfate has marked effects on the stability and solubility of barnase[33], whereas the osmolytes have rather small effects[78]. If the Hofmeister effect is explained simply from the indirect effects of co-solutes on water, then how can this difference be explained? We propose that the difference is due to the binding of Hofmeister ions to the protein, which (obviously but non-trivially) leads to a change in the surface charge of the protein. In colloid science, the stability of a colloid (that is, its tendency to flocculate) is described by the zeta potential: the electrical charge inside the interface between the particle and the solution. In the same way, the binding of a Hofmeister ion to a protein changes the surface potential of the protein, and therefore has major effects on the behavior of the protein in solution. Because the binding of Hofmeister ions to proteins saturates by about 100–200 mM, but the Hofmeister effects generally only start to be apparent at 100 mM and continue to 1 M and beyond, it is necessarily the case that almost all the ion binding must be complete before Hofmeister effects are significant. This means that one cannot isolate the charge effect from the

Hofmeister effect—except of course by using an uncharged kosmotropic osmolyte, which (as predicted by our explanation) stabilizes the protein without much effect on the solubility.

In our consideration of titrations with pairs of Hofmeister ions, we saw that the binding of one ion changes the fluctuations at the protein/water interface, and therefore strengthens the binding of the second ion. This effect is likely to provide a further difference between Hofmeister ions and osmolytes: titration with a Hofmeister ion results in binding, which necessarily directly reduces the fluctuations at the protein/water interface, in a way that osmolytes do not do.

This discussion also provides a simple rationale for the inverse Hofmeister effect, which is seen for proteins below their pI, in other words for proteins with net positive charge. For such proteins, titration with a negatively charged Hofmeister anion leads initially to a reduction in overall charge, until such point as the total protein charge becomes negative, after which further addition of anions produces an increase in negative charge, as it does for proteins above their pI. Hence, inverse Hofmeister effects are essentially due to the protein/ion complex passing through its pI; once it has done this, then it behaves in a normal Hofmeister manner. We are by no means the first to propose ideas similar to this[13,38,79], and the effect clearly involves more than simply charge, but we propose that surface charge provides a simple and largely sufficient rationalization.

Ions on the left of the Hofmeister series (kosmotropes, such as sulfate or phosphate) stabilize proteins but make them soluble, whereas ions on the right do the opposite. Typically what one is looking for in an excipient is something that makes the protein both more soluble and more stable. Is this possible—that is, are the effects of Hofmeister ions on stability and solubility two aspects of the same mechanism, or is it possible to find a suitable combination of excipients that achieves both solubility and stability at the same time? The discussion above implies that these two aspects are in principle separable, because there are two different principles at work: solvent fluctuations (which we can quantify using $m_L$), and ion binding/surface charge. There are two other properties that we have not explored here: one is co-solute size (via the volume exclusion effect) and the other is site-specific binding, which is probably promoted by large charge-diffuse ions[14]. Both of these could also be exploited to achieve both greater stability and greater solubility.

To summarize, we propose that Hofmeister effects can be best understood as an effect of kosmotropes vs chaotropes; however, kosmotropes should be understood to act by restricting the fluctuations of water, while chaotropes promote fluctuations (Fig. 10). These fluctuations occur throughout the solvent including at the solvent/protein interface. By restricting fluctuations, kosmotropes promote the enthalpic effect of hydrogen bonding and therefore stabilize proteins, while acting to salt them out and therefore make them less soluble. Chaotropes have the opposite effect. Hofmeister ions also bind to proteins, thereby making them more negatively charged. For proteins below their pI, this effect leads to proteins passing through their pI, which is the origin of the inverse Hofmeister effect. The binding of Hofmeister ions to the protein also serves to reduce solvent fluctuations at the protein surface. By contrast, osmolytes behave as kosmotropes but do not bind to the protein surface; and in addition both osmolytes and Hofmeister ions have an intrinsic stabilizing effect arising from simple excluded volume. These hypotheses are undoubtedly simplistic[80], but have the merit that they can easily be tested via Kirkwood-Buff theory, which provides a link between the thermodynamic parameters and simple physical measurements, such as neutron or X-ray diffraction, or compressibility, and density.

## Methods
### Experimental details
The expression and purification of barnase were carried out as described[33,81]. Briefly, the catalytically inactive H102A mutant was carried on a pQE60 plasmid, expressed in *Escherichia coli* M15 [pRep4] cells in M9 minimal medium with uniform [13]C/[15]N labeling, and purified using Q-sepharose and SP-sepharose columns. Protein was dialyzed extensively against HPLC

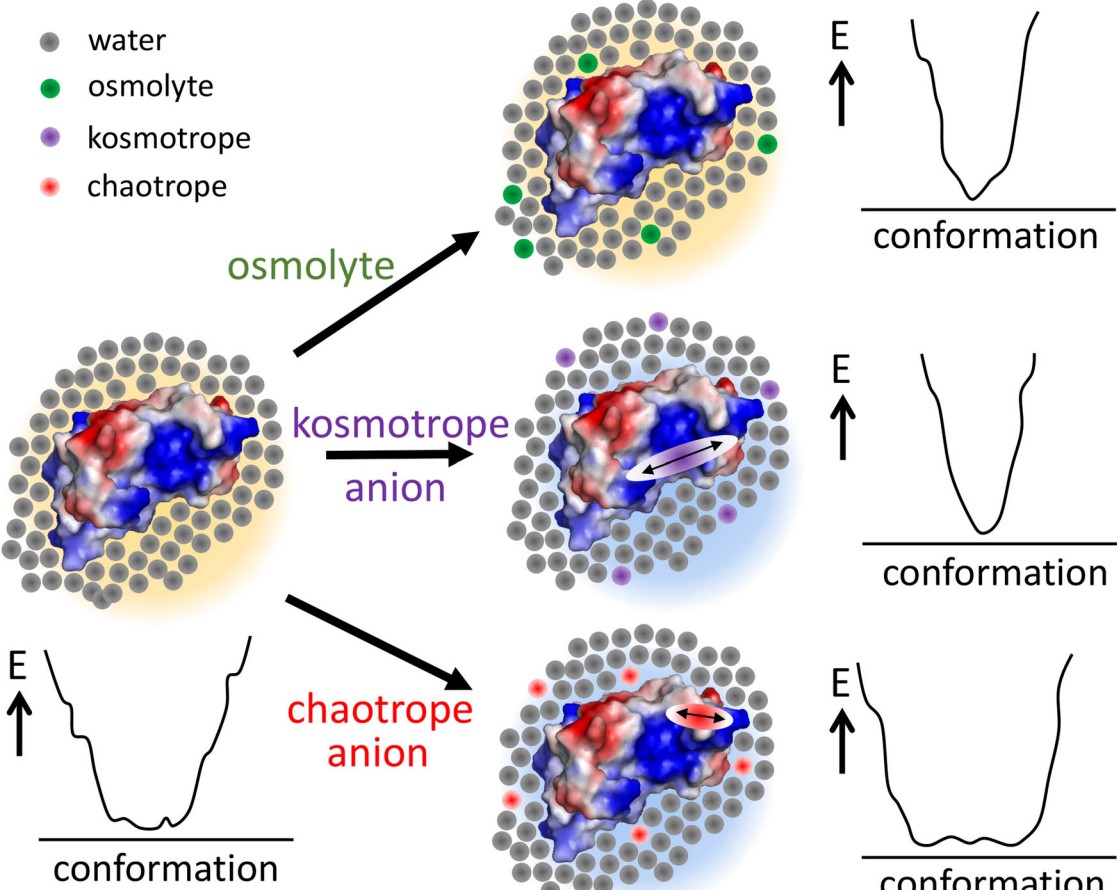

**Fig. 10 | Summary of key conclusions.** Barnase is shown as an electrostatic surface (blue=positive, with the active site at center right), surrounded by solvent. The energy well represents the extent of fluctuations within the solvent (and because of slaving, also within the protein). Added osmolyte is mainly excluded from the protein surface and leads to a reduction in fluctuations which stabilizes the protein. Added Hofmeister kosmotropic anion (eg sulfate) similarly is excluded and produces a reduction in fluctuations; in addition, it binds to the protein surface and so changes the overall change on the protein (background color). Added chaotropic anion (eg thiocyanate) is less excluded and also binds to the protein surface, producing a change in charge and an increase in fluctuations.

grade water. All experiments were carried out in 5 mM sodium acetate, pH 5.8, typically with 500 μM barnase in 10% $D_2O$. Most NMR experiments were carried out using a shaped tube to avoid loss of sensitivity at high salt concentrations. Co-solutes were prepared as stock solutions of 1 – 3 M in the same buffer. For titrations with pairs of ions, the two ions were mixed together in a 1:1 ratio and used as a stock solution, with 1–2 M of each. To avoid handling errors, test titrations were carried out prior to the NMR titration and the pH was followed. Where necessary, the pH of the co-solute stock was altered slightly to avoid pH changes larger than 0.1 pH units during the titration. This procedure has the disadvantage that there are small pH changes during the titration, but has the big advantage of less handling and therefore much more consistent volumes and concentrations[82].

### NMR methods

Two-dimensional $^{15}N$ HSQC, HNCO, and $^{13}C$ HSQC spectra were obtained at each titration point at 298 K on a Bruker DRX-600 with a cryoprobe. Spectra were referenced to TSP and exported to Felix (Felix NMR, Inc., San Diego, CA) for peak picking into a database. Data were fitted to Eq. (3) using Levenberg–Marquardt least squares fitting. The NMR assignments for barnase were obtained from BMRB[83] entry 4964[84] and checked. X-ray coordinates for barnase were taken from the pdb[85] file 1a2p[86].

Following extensive tests, the following procedure was followed in order to obtain confident fits. All datasets were fitted initially to Eq. (3). Very weak binding can produce curves with slight curvature, which fit to weak affinities and very large $\Delta\delta_{max}$; because the slope is fit as a binding

and not a linear slope, the magnitude of the $m_L$ is then erroneously small. These fits were eliminated by fitting to a simple linear gradient if the fitted $K_d$ was larger than 900 mM. It was also sometimes observed that the first addition of co-solute gave a small chemical shift change, with essentially no further change in the shift on subsequent additions. This behavior fits to very tight (but meaningless) affinities, and was avoided by ignoring any nuclei fitted to a $K_d$ stronger than 5 mM. Finally, for any nuclei that fit with an absolute $\Delta\delta_{max}$ of less than 0.03 ppm ($^1H$) or 0.06 ppm ($^{13}C$ and $^{15}N$), the fitted binding was ignored and the data were fit to a simple linear gradient. The fits to Eq. (3) and to linear gradients are listed in Supplementary Data 1.

### Statistics

Statistical comparisons were done using Microsoft Excel®. Protein structures were examined using Pymol (Schrödinger, Inc.). The solvent accessible area was calculated using the program Naccess (S. Hubbard, University of Manchester, UK). The figures showing structures colored by association constant (Figs. 2, 4) were produced as putty plots in Pymol. The colors were inserted into the $B$ factor column of the PDB file using a Python script, such that all atoms in each residue had a $B$ value calculated as the log of the averaged $K_a$, scaled to generate $B$ values between 0 and 10.

### Reporting summary

Further information on research design is available in the Nature Portfolio Reporting Summary linked to this article.

## Data availability
Supplementary Data 1 contains a complete table of fitted $K_d$, $\Delta\delta_{max}$ and $m_L$ for each co-solute binding to barnase, fitted for $^{15}N$, $^1H_N$ and $^{13}C'$, example sets of data for shift changes of methyl $^{13}C$ and $^1H$ on binding of TMAO, and changes of $^1H$, $^{15}N$ and $^{13}C$ on binding of a 1:1 mixture of thiocyanate and sulfate, a figure showing the hydrogen bonding network around L42 and R83, and a figure showing residue numbering in barnase. The authors declare that the data supporting the findings of this study are available within the paper and its Supplementary Information and Supplementary Data 1 files. Should any raw data files be needed in another format they are available from the corresponding author upon reasonable request.

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

## Acknowledgements

The authors thank BBSRC (BB/P007066/1; BB/P020038/1) and the University of Sheffield for financial support (C.R.T., N.J.F. and Y.K.D.R., respectively), and Andrea Hounslow for assistance with the NMR experiments. We thank the anonymous referees for their perceptive comments.

## Author contributions

Methodology: C.R.T. and N.J.F.; investigation: C.R.T. and Y.K.D.R.; writing—original draft: C.R.T., Y.K.D.R. and M.P.W.; writing—review and editing; M.P.W.; supervision: M.P.W.; funding acquisition: M.P.W.

## Competing interests

The authors declare no competing interests.
