## [Peer Review File · Communications Chemistry]

Reviewers' comments:

Reviewer #1 (Remarks to the Author):

The manuscript by Trevitt et al reports an NMR chemical shift titration study on the site-specific interactions of salts and osmolytes with the protein barnase in solution. The solute dependent chemical shifts are modelled assuming specific association and a medium effect. The authors find regions of barnase where anions bind and experiments with mixed salt solutions confirm additive interactions with respect to the ions' binding sites, yet the interaction strengths are not fully additive. For osmolytes, site-specific binding strengths are weaker and the effect of the protein appears to be dominated by the medium effect. Correlation of osmolyte effects for different osmolytes at the same protein sites suggests that osmolytes indirectly affect the protein.

I find the study highly interesting, the data are of high quality, and the analysis has been thoroughly performed. The study offers very detailed insights into the question how solutes and salts affect proteins. I have a few comments on the interpretation of the results and some minor details on the data analysis should be elaborated. If the authors can address these comments, I recommend publication of the manuscript in Communications Chemistry.

- My main concern is about the interpretation of the binding constants. After reading the manuscript, I got the impression that the authors do not consider solute-induced changes to the conformation of the protein. Alternative to the direct binding of ions to the protein giving rise to the variation in chemical shift, also a salt induced change to the protein conformation (without the ions directly binding to the protein) could alter the chemical shift. For the latter case, the sites of the protein with the most pronounced conformational changes would show the most intense changes in the shift, while binding of the ions may have occurred at a different site. This seems to be in line with association of ions to small peptides being much weaker (doi 10.1038/s42004-022-00789-y, 10.1021/jp405683s) than for proteins like reported here (or e.g. also 10.1038/s41598-023-29901-5). In a recent paper (10.1038/s41598-023-29901-5) this difference has been ascribed to curvature, but the authors of this study also discuss conformational changes in the SI.

Additionally, conformational changes may contribute to the medium effect (m_L) for motionally averaged NMR signals, like demonstrated here: 10.1021/acs.jpcc.6b12465.

I understand that it is very difficult to isolate conformational effects from the reported results, yet a much more thorough discussion of potential effects of conformational changes should be included.

- Some of the authors have published partially overlapping study on the same combination of protein and salt (Ref 63, 10.1021/acsomega.6b00223). At first glance, the conclusions of the current manuscript seem to differ from the conclusions of the earlier study. The authors should explain the advance/difference with respect to this earlier work and how/why the present findings are consistent/or inconsistent with the earlier work.

- From the caption of Fig 1 it is not clear, which curve corresponds to which salt (or salt mixture)? This should be clarified.

- It is not clear to me how the association constants in Fig 4 have been determined for the mixed salts

(based on total salt concentration or based on the concentration of one of the salts). I presume the total salt concentration. If this is the case, the enhanced association for mixed salts is even more surprising, as the concentration of the 'binding anion' is only half of the salt concentration. I think, this needs a more thorough discussion.

- The meaning of the distributions in Fig 5d is not very clear to me and needs more explanations. The authors argue that these values are additive for the mixed salts, yet, as far as I understand, for additivity the values of the mixed salts should be just the average of the values for the individual salts, which seems to be not supported by this figure. More explanations are needed here.

- The authors discuss the origins of synergistic effects for mixed salts on page 16 in terms of dynamic fluctuations. There has been a different mechanism suggested recently (10.1021/jacs.9b00295), which should be considered as an alternative explanation.

- The authors interpret their findings in terms of solvent fluctuations giving rise to specific ion effects. While this could be true, there is little direct evidence for this mechanism presented in the paper. It would strengthen the paper, if the authors could make it more explicit that this is just a plausible explanation, yet, rather speculative. Ideally, the authors outline possible experiments/simulations on how to test their hypothesis.

Reviewer #2 (Remarks to the Author):

This article applies NMR for measuring weak interactions between salts and osmolytes around a small protein barnase. The types of interactions can be grouped according to the binding behaviour, where the stronger interactions can be described by a binding model and have affinities with dissociation constants around 1 to 100 mM, while the "ultra-weak" interactions are reflected by a linear change in the chemical shifts over the whole concentration range. While there exist a few studies determining ion or excipient binding with affinities around 10 mM, the "direct" measurement of the "ultra-weak" interactions by NMR or by any other spectroscopic method have not been reported before. These are the types of interactions that dictate the effects of co-solvents on protein solubility or stability when at co-solvent concentrations exceeding 100 mM or more. Previously these solvent-protein interactions have only been accessible in terms of parameters reflecting averaged interactions over the entire protein surface, while here, the interactions are determined about each amino acid group. As such, this work will surely lead to new and important insights about solvent interactions with proteins, which are especially significant since these interaction dictate solubility and stability of protein solutions.

I have some minor comments for clarifying a few important points in the manuscript.

I found the discussion of different theories about protein-solvent interactions on pages 70 to 77 confusing. I am not sure what is meant by the preferential interaction theory? References 16 and 17 correspond to reviews by Timasheff, who measured preferential interaction parameters between co-solvents and proteins and used these to determine how the co-solvents altered protein stability and

solubility according to thermodynamic linkage phenomena. The theory of preferential interactions relates the averaged molecular interactions between co-solvents and proteins (eg preferential interaction parameters) to how co-solvents alter the protein free energy, but the theory does not provide the molecular basis for these interactions. In Timasheff's work he found that chaotropic anions such as SCN^- exhibit positive preferential interaction parameters with proteins, but did not attribute the binding to interactions with hydrophobic groups. Rather the positive preferential interaction parameters for chaotropic anions were rationalized in terms of a balance between the preferential exclusion of the anion about the hydrophobic regions of the protein and preferential binding interactions that overcome these exclusion interactions. As such, this section would benefit from a more clear definition of what is meant by preferential interaction theory versus preferential hydration theory.

I also found the statement that it is not clear which of the three theories is best, a bit confusing. The three different theories could all be applicable and contribute to the interactions of co-solvents with proteins. As it is written, the preferential interaction theory describes the interaction with non-polar groups, while the preferential hydration theory describes the interaction with polar groups. The volume exclusion mechanism provides just one contribution to the interaction of the co-solvent with either type of these groups. The volume exclusion mechanism would dominate if the co-solvent was very big compared to water, and has been used to describe the effect of polymers on protein stability, and to a lesser extent the effects of big osmolytes such as sugars. So any unifying approach will likely require including all these mechanisms since proteins contain both polar and non-polar groups. That ion interactions with polar versus non-polar groups differ from each other is known from salting-in and salting-out studies of model peptides done back in the 1960 and 1970s. In these studies, model peptides were capped and did not contain any charged groups so that the peptide free energy (as determined from their solubility or partitioning studies) was linear with respect to salt concentration over the whole concentration range (eg only the ultra-weak interactions dictate the salting-in or salting-out behaviour). These studies were used by Record to quantify the anion and cation interactions with polar versus non-polar groups (see Figure 13 of reference 18). Record did find that chaotropic anions partition or preferentially bind to non-polar surfaces albeit very weakly, while kosmotropic anions are excluded from about these surfaces, which was then confirmed by the simulations referred to in reference 19. I think the paragraph starting on line 65 would benefit by making a more clear distinction about these differences. While Record results need to be tested since they are based on many simplifying assumptions (eg additivity and the short-ranged nature of co-solvent protein interactions), they do illustrate a key point that ultra-weak interactions, which are reflected by the linear slopes of chemical shifts (mL) presented here, are due to competing effects of preferential partitioning and preferential exclusion interactions. Indeed, for any destabilizing co-solvent, such as urea and to a lesser extent SCN^- , the net effect is preferential partitioning around the protein, that is, the local concentration of the denaturant about the protein surface must be greater than the bulk concentration of the denaturant.

Following on from the above points, I am confused by the section on lines 266 to 272. It is not clear to me how the authors have deduced that there is no greater binding of chaotropes to hydrophobic regions than of kosmotropes. The mL value does follow the Hofmeister series suggesting that there is at least a difference in the solvent-mediated interactions, although it is not clear if this arises due to solvent-mediated interactions about polar groups or about non-polar groups?

In Figure 9, I think it would be insightful to compare sulfate and chloride with either ectoine or betaine, and then to compare sulfate/chloride to SCN⁻. A strong correlation indicates that the solvent mediated interaction is only determined by the interaction of the co-solvent with water and independent of the chemical nature of the protein surface.

On line 502, I would suggest the following clarification.. : "any co-solute added to a protein will increase protein stability if the co-solute does not preferentially interact with the protein", note that denaturing co-solutes (eg urea) decrease protein stability because weak binding interactions overcome the volume exclusion effect.

On line 514, there is a mis-understanding about the meaning of fluctuations. KB integrals relate to density fluctuations of the solvent and co-solvent molecules, which are "static" properties, and do not relate to any dynamic property (eg folding rate constants, diffusion coefficients, conformational dynamics).

Paragraph starting on line 523 - As far as I am aware, most approaches applying KB theory treat the protein as a rigid particle so it is not clear how protein fluctuations would relate to the solvent and co-solvent density fluctuations.

Reviewer #3 (Remarks to the Author):

This paper discusses the relationship between protein stability and the state of water influenced by some ions with different properties, based on NMR chemical shift data. Whereas the chemical shift perturbations observed in typical NMR titration experiments are generally expressed as a hyperbolic function, this study observed linear chemical shift perturbation patterns in addition to a curvature change that converges to $\delta\Delta_{max}$ at a certain titration point. Therefore, the authors introduced an additional term with a new variable, mL , to elucidate this phenomenon, in conjunction with the hyperbolic function including a variable for binding affinity (K_d or K_a).

In this study, the authors conducted NMR titration experiments with barnase and several representative Hofmeister ions and osmolytes, exploring the interactions between ions and proteins, and water states. They explained the effects and roles of these ions in protein stabilisation and destabilisation. This research is a very unique and interesting, as few studies have undertaken such an analysis based on NMR titration experiment data. This paper is evaluated as an original and significant research work, meriting publication in communications chemistry.

However, given its novelty and bold hypotheses, the paper warrants cautious discussion. A major issue is that the discussion relies almost exclusively on chemical shift perturbation data from NMR experiments. While the data analyses are robust, the paper would be more convincing if the following points were addressed and corrected.

1. The authors explain all observed chemical shift perturbations exclusively by changes of hydrogen bonds. Generally, chemical shifts are influenced by various other factors and do not directly correlate

with simple hydrogen bond strength alone. The authors should describe other possibilities or reasons why the authors eliminated them.

2. The authors introduce a new variable, mL , to account for indirect effects leading to chemical shift perturbations. These indirect effects encompass changes in solvation other than intermolecular interactions, predominantly focusing on hydrogen bond strength. While the calculated mL values correlate with hydrogen bond strength, the authors do not distinguish between hydrogen bonds formed with water molecules on the molecular surface and those within internal molecular atoms. This lack of distinction raises questions about the validity of treating these different types of hydrogen bonding equivalently. Specifically, amide protons on the molecular surface are in continual exchange with hydrogen atoms in water, in addition to forming hydrogen bonds. Considering the timescales of chemical shift observations and the hydrogen atom exchange rate, is it justifiable to overlook hydrogen atom exchange and other effects? Moreover, given that changes in solutes also impact chemical shift perturbations inside the molecule, it implies that the structure and dynamics of the entire molecule are affected to some extent. Is it accurate to attribute these changes solely to variations in hydrogen bond strength? Should the analysis also consider other factors influencing chemical shifts, such as ring-current effects?

3. The authors assert that only a single ion binds to the protein surface, as indicated by K_d values, and that the fluctuation of this ion on the surface leads to chemical shift changes. This claim prompts a critical inquiry into the timescale of ion fluctuation on the surface. How does this timescale align with the timescale over which fast-exchange chemical shift changes are observed? The paper mentions that the results of ion fluctuation on the surface are replicated by MD simulations, etc. It would be beneficial for the authors to elucidate the correspondence of these timescales with those known from MD simulations, to reinforce the validity of their findings.

4. The discussion section of the paper is extensive and complex that could hinder comprehension, especially due to its text-heavy format. Considering that *Communications Chemistry* is a general chemistry journal, not specifically focused on physical chemistry, and that its audience includes biochemistry researchers, it is crucial to present more concise and easily understandable explanations. The incorporation of several schematic diagrams would significantly aid in visually conveying the authors' concepts, thereby improving the paper's overall clarity. Additionally, the manuscript would be strengthened by the explicit inclusion of the physico-chemical equations that form the basis of the discussion, which are currently only described textually. Accompanying these equations with clear, explanatory notes would render the content more accessible and engaging for a wider readership.

The following minor points should also be addressed:

1. p3 L39: In the manuscript up to this point, there is a detailed explanation of kosmotropes and chaotropes. However, there is no definition for Hofmeister anions and osmolytes, and no explanation regarding their relationship between them, and these terms suddenly appear here. It is essential to include a clear and concise explanation of Hofmeister anions and osmolytes, particularly elucidating their relationship with kosmotropes and chaotropes, to avoid confusion and ensure clarity for readers

not familiar with these terms.

2. p3 L119: "while the linear parts represent indirect effects on the protein chemical shifts caused by the interactions of the co-solute with the solvent, which thus lead to altered protein solvation". It is mentioned that the linear chemical shift change in mL of the slope continues at least up to 1M. However, it seems implausible that this trend would continue indefinitely. It is reasonable to expect that the change will eventually reach a saturation point where the slope alters. The manuscript should address this aspect: up to what point can this linear relationship be considered valid? Clarification on the extent to which this theory can be applied is necessary to understand the limitations and applicability of these findings.

3. p5 L164: "Almost the only generalisation that one can make about chemical shift changes is to say that a downfield shift of an amide proton indicates an increase in the strength of hydrogen bonding involving that proton." It is noted that the discussion up to this point has been based on the data presented in Figure 1. However, Figure 1 illustrates the chemical shift of ^{15}N . This raises a question: why have the authors chosen not to utilize ^1H chemical shifts in their analysis? The manuscript should provide an explanation or justification for the use of ^{15}N chemical shift data in the context of discussing hydrogen-related phenomena.

4. Equation (1): The variable 'i' is used but not clearly defined. Clearly specify it.

5. Figure 2: The manuscript or figure should specify the criteria used for the color changes mapped onto the structure. Additionally, the text discusses certain amino acid residues (35-46, 81-86), but their locations are not clearly identifiable in the structure depicted in the figure. To aid reader comprehension, please either label the amino acid numbers directly on the structure, or number the secondary structures ($\alpha 1$, $\alpha 2$, $\beta 1$, etc.) and clearly indicate which regions are being discussed.

6. p7 L215: 'almost entirely linear.'. It is unclear which specific data or part of the analysis this statement refers to. For clarity and accuracy, it is essential that the manuscript explicitly specify the exact data or results being described as 'almost entirely linear.'

7. p7 L225: "This region of the protein is not the most positively charged patch (which is the active site of this RNase), but it is close to the active site and to other positively charged regions.". The manuscript lacks a diagram of the electrostatic potential map on the 3D structure, making it difficult to identify charge locations. The manuscript should include such a diagram.

8. p7 L235, "residues 27-31 and 54-62": Not sure which area in Figure 2c. Please indicate clearly in the Figure.

9. p8 L264: "Further support for this view comes from the fact that Hofmeister effects are widely held to be directly proportional to co-solute concentration, up to and beyond 1 M." References should be given.

10. p8 L270: "We therefore rule out the preferential interaction theory, and also the preferential hydration theory, which leaves as possible explanations something like the original

chaotrope/kosmotrope idea (ie for example that kosmotropes withdraw water from the protein surface) and the volume excluded theory, which has many elements in common with the chaotrope/kosmotrope idea." "preferential interaction theory", "preferential hydration theory", " The terms "preferential interaction theory", "preferential hydration theory", "original chaotrope/kosmotrope idea", etc. are not explained, and the point is not clear. It would be better to move these sentences to the discussion section and give more detailed explanations.

11. p8 L277: "A positive 1HN mL (as found for the kosmotrope sulfate) implies that titration with sulfate increases the strength of hydrogen bonding from the surface of the protein to water." For residues exhibiting negative mL values, which represent buried amide protons, discussing hydrogen bond with water seems incongruous. Could the authors clarify their explanation for this scenario?

12. p9 L290: "bulk water is more ordered in the presence of kosmotropes." Could the authors clarify how bulk water, in its ordered state, is related to the process of drawing hydration water from the protein surface?

13. p9 l290: "This would lead us to expect that buried amides would have mL with an opposite sign to the mL of surface-exposed amides. ." In Figure 3, there doesn't seem to be a distinct trend distinguishing positive mL values from negative ones. Could the authors specify the exact region of change they are referring to in this statement? Additionally, in the sentence above, the manuscript mentions, "proteins are very complex systems with dynamics". Yet, it concludes that chemical shift perturbations imply 'increasing the strength of interactions elsewhere (for example, inside the protein).' How can the authors assert that the interactions within the protein, presumably hydrogen bonds, have become stronger based solely on chemical shift perturbations?

14. p15 L469: "the binding of an ion, even when that binding is spread over a large surface patch, reduces dynamic fluctuations at the surface. " The logic of this and the subsequent statement is unclear. Based on the description provided, the binding of an ion to a protein surface seems to involve the ion moving around on the protein surface. How does this movement clearly suppress protein surface fluctuations? Is it possible to conclusively investigate such dynamics, for instance, through NMR relaxation analysis? Additionally, the manuscript should clarify the experimental basis for the assertion that suppressing protein fluctuations aids in the binding of a second ion. It is commonly believed that greater flexibility in a protein's binding surface allows for more diverse binding. Does this principle not apply to ions?

15. p16 L520: "KB theory emphasises Kirkwood-Buff integrals ..." Please describe the formula and explain it clearly. It is not clear from the text alone.

16. p16 L520: "Lower solvent fluctuations reduce the ability of the solvent to hydrate the protein surface" I did not understand the logic behind this.

17. Method section should state what stable isotope labelling was done - ^{13}C , ^{15}N uniform label sample?

18. Figures S2 and S3, which show profiles of chemical shift changes and fitting, are not cited anywhere in the text. Please ensure these figures are referenced in relevant sections. This will provide clarity and support for the claim 'no significant curvature' in the titration of osmolytes, helping the reader to understand and verify this assertion.

19. In the introduction, the authors express an intent to identify effective excipients for protein stabilization. It would be beneficial for the biological readership if the manuscript concluded with brief guidance or recommendations on which excipients to choose based on the study's findings. This addition would provide practical value and aid in the application of the research.

Dear Reviewers,

Thank you for sending the referees' comments on the paper. The referees have made many helpful comments, which have significantly improved the paper. We have made detailed and extensive changes to the paper in response. I am attaching a revised version of the paper with revisions marked in red. In detail:

Reviewer 1

1. *My main concern is about the interpretation of the binding constants. After reading the manuscript, I got the impression that the authors do not consider solute-induced changes to the conformation of the protein. Alternative to the direct binding of ions to the protein giving rise to the variation in chemical shift, also a salt induced change to the protein conformation (without the ions directly binding to the protein) could alter the chemical shift. For the latter case, the sites of the protein with the most pronounced conformational changes would show the most intense changes in the shift, while binding of the ions may have occurred at a different site. We agree that conformational changes would alter the chemical shift, and that the largest shift changes would indicate the largest conformational change. However, we suggest that there is no difference (thermodynamically or structurally) between a change in solvation and a structural change arising from altered solvation. A change in solvation of the protein will only alter chemical shifts if there is some kind of structural change (including possibly hydrogen bonds between protein and solvent). Fundamentally this is a question about binding affinities. In order to see a curvature of chemical shift vs co-solute by NMR, we require a binding affinity in the mM range (or stronger). Solvent-mediated interactions are generally much weaker than this (1 M dissociation constant or weaker), and cause shift changes that are effectively linear with co-solute concentration. Schellman made a similar point (expressed in thermodynamic terms) when he pointed out that very weak co-solute/protein interactions are best considered as a change in solvation. We have added these points to the manuscript (lines 142-147).*

This seems to be in line with association of ions to small peptides being much weaker than for proteins like reported here (references). We agree. We have included citations to some of these references where relevant at different places. We note that doi 10.1038/s41598-023-29901-5 refers to non-additive effects of combinations of ions, a topic that we also discuss. However, the two set of non-additive effects are completely different and we felt it would be confusing to refer to this paper at this point in our discussion (though we do discuss it elsewhere).

Additionally, conformational changes may contribute to the medium effect (m_L) for motionally averaged NMR signals. We agree. This is the same point that we made above – conformational change and medium effect are two aspects of the same phenomenon.

I understand that it is very difficult to isolate conformational effects from the reported results, yet a much more thorough discussion of potential effects of conformational changes should be included. I hope our additional text has satisfied the referee. Overall, the chemical shift changes are small and clearly show that there is no major conformational change on addition of co-solutes. Rather, the protein adjusts to changes in solvation by (for example) small alterations to hydrogen bonding, both to solvent and internally.

2. *Some of the authors have published partially overlapping study on the same combination of protein and salt (Ref 63, 10.1021/acsomega.6b00223). At first glance, the conclusions of the current manuscript seem to differ from the conclusions of the earlier study. The authors should explain the advance/difference with respect to this earlier work and how/why the present*

findings are consistent/or inconsistent with the earlier work. Thank you for pointing this out. We agree that we should have explicitly compared the present results to this earlier paper, which we now do (lines 111-115).

3. *From the caption of Fig 1 it is not clear, which curve corresponds to which salt (or salt mixture)? This should be clarified.* Thank you: these have now been clarified.
4. *It is not clear to me how the association constants in Fig 4 have been determined for the mixed salts (based on total salt concentration or based on the concentration of one of the salts). I presume the total salt concentration. If this is the case, the enhanced association for mixed salts is even more surprising, as the concentration of the 'binding anion' is only half of the salt concentration. I think, this needs a more thorough discussion.* Thank you: this is an important point. We have added text (lines 340-344) to clarify that the ion concentrations are based on the (equal) concentrations of each ion, rather than their sum (so not the total salt concentration). As noted in the new text, this means that if there is no interaction between the ions, and no change in protein structure or dynamics when one ion binds, then one would expect the affinities for single ions and pairs of ions to be identical. The fact that they are not is the basis for our argument (lines 509-515) that binding of one ion must restrict protein fluctuations. In light of the fact that it is concentration of each ion, not total ion concentration, we feel that the current discussion (lines 509-515) is appropriate, and that further discussion would be over-speculative.
5. *The meaning of the distributions in Fig 5d is not very clear to me and needs more explanations. The authors argue that these values are additive for the mixed salts, yet, as far as I understand, for additivity the values of the mixed salts should be just the average of the values for the individual salts, which seems to be not supported by this figure. More explanations are needed here.* For the same reasons as given in #4 above, the values for mixed salts are expected to be the sums, not the averages, of the two individual ions. Within experimental error, they are: the least good is thiocyanate + sulfate, which also has the largest error. We have added additional text (lines 373-375) to provide more explanation. We should add that the figure panels were referenced wrongly in the original text, which did not help: these have now been corrected.
6. *The authors discuss the origins of synergistic effects for mixed salts on page 16 in terms of dynamic fluctuations. There has been a different mechanism suggested recently (10.1021/jacs.9b00295), which should be considered as an alternative explanation.* Thank you for pointing this out: we have added new text (lines 516-524) to discuss this useful suggestion.
7. *The authors interpret their findings in terms of solvent fluctuations giving rise to specific ion effects. While this could be true, there is little direct evidence for this mechanism presented in the paper. It would strengthen the paper, if the authors could make it more explicit that this is just a plausible explanation, yet, rather speculative. Ideally, the authors outline possible experiments/simulations on how to test their hypothesis.* In our original text, we had attempted to do exactly this (lines 595-601 of the revised text). Clearly this was an inadequate discussion. We have therefore added further details; in particular, to make it clear that this is not a question of alternative mechanisms (fluctuations OR structure OR solvent withdrawal), but rather a question of the language used to describe the same biophysical events. I am not sure how helpful this is, but the revised text is hinting at this being a 'paradigm shift' as discussed by Kuhn: not so much a different mechanism, but a different point of view.

Reviewer 2

We thank the reviewer for their perceptive and sympathetic summary of our argument, and for the high level of attention they have devoted to this review.

1. *I found the discussion of different theories about protein-solvent interactions on [lines] 70 to 77 confusing. I am not sure what is meant by the preferential interaction theory? References 16 and 17 correspond to reviews by Timasheff, who measured preferential interaction parameters between co-solvents and proteins and used these to determine how the co-solvents altered protein stability and solubility according to thermodynamic linkage phenomena. The theory of preferential interactions relates the averaged molecular interactions between co-solvents and proteins (eg preferential interaction parameters) to how co-solvents alter the protein free energy, but the theory does not provide the molecular basis for these interactions. In Timasheff's work he found that chaotropic anions such as SCN⁻ exhibit positive preferential interaction parameters with proteins, but did not attribute the binding to interactions with hydrophobic groups. Rather the positive preferential interaction parameters for chaotropic anions were rationalized in terms of a balance between the preferential exclusion of the anion about the hydrophobic regions of the protein and preferential binding interactions that overcome these exclusion interactions. As such, this section would benefit from a more clear definition of what is meant by preferential interaction theory versus preferential hydration theory. Thank you for the helpful clarification: we agree that this was a confusing discussion. We have accordingly rewritten the text (lines 69-93) to make this clearer and more accurate. As part of the change, we have removed the distinction between preferential hydration and preferential interaction, which we agree is not a helpful comparison.*
2. *I also found the statement that it is not clear which of the three theories is best, a bit confusing. The three different theories could all be applicable and contribute to the interactions of co-solvents with proteins. As it is written, the preferential interaction theory describes the interaction with non-polar groups, while the preferential hydration theory describes the interaction with polar groups. The volume exclusion mechanism provides just one contribution to the interaction of the co-solvent with either type of these groups. The volume exclusion mechanism would dominate if the co-solvent was very big compared to water, and has been used to describe the effect of polymers on protein stability, and to a lesser extent the effects of big osmolytes such as sugars. So any unifying approach will likely require including all these mechanisms since proteins contain both polar and non-polar groups. That ion interactions with polar versus non-polar groups differ from each other is known from salting-in and salting-out studies of model peptides done back in the 1960 and 1970s. In these studies, model peptides were capped and did not contain any charged groups so that the peptide free energy (as determined from their solubility or partitioning studies) was linear with respect to salt concentration over the whole concentration range (eg only the ultra-weak interactions dictate the salting-in or salting-out behaviour). These studies were used by Record to quantify the anion and cation interactions with polar versus non-polar groups (see Figure 13 of reference 18). Record did find that chaotropic anions partition or preferentially bind to non-polar surfaces albeit very weakly, while kosmotropic anions are excluded from about these surfaces, which was then confirmed by the simulations referred to in reference 19. I think the paragraph starting on line 65 would benefit by making a more clear distinction about these differences. While Record results need to be tested since they are based on many simplifying assumptions (eg additivity and the short-ranged nature of co-solvent protein interactions), they do illustrate a key point that ultra-weak interactions, which are reflected by the linear slopes of chemical shifts (δ) presented here, are due to competing effects of preferential partitioning and preferential exclusion interactions. Indeed, for any destabilizing co-solvent, such as urea and to a lesser extent SCN⁻, the net effect is preferential partitioning around the protein, that is, the local concentration of the*

denaturant about the protein surface must be greater than the bulk concentration of the denaturant. This is a helpful and detailed point. I agree with almost all of it, my only point of contention being the final argument over preferential partitioning of destabilising co-solutes to the protein surface. There is good evidence that urea binds to the protein surface, and that this binding is a major contributor to its stabilising effect; but the evidence for preferential binding of chaotropes such as iodide or thiocyanate is less compelling. For example, Rankin doi.org/10.1021/ja4036303 argues that iodide does enter the hydrophobic hydration shell but does not really bind to the protein surface; it just is less strongly expelled than more hydrated ions such as fluoride. This is related to the point now added at lines 142-147 in response to referee 1 (#1) that very weak binding can be treated as a change in solvation rather than as a specific binding interaction. Our major conclusion in this paper is that the effects of co-solutes are best understood as changes to the fluctuations in the system (solvent and protein, and their interactions). We point towards an explanation based on Kirkwood-Buff theory, which connects structural and thermodynamic aspects: in other words, that an understanding of co-solute effects should not focus exclusively on structure or on thermodynamics but needs to consider both (possibly by phrasing the discussion in terms of fluctuations). Hence we agree with the referee that to ask which of these theories is best, is not a helpful approach. This is essentially the point made in the final Discussion section of the paper, and we suggest that the points raised by the referee in this comment are best taken up in the discussion and not in this introduction, where they would introduce needless complication. Accordingly, we have added a short paragraph in the introduction (lines 87-93), pointing towards the discussion. Finally, we note that a consequence of our emphasis on the dynamic effects of co-solutes is that the detailed structural interactions between solutes and the protein surface become less important. We therefore propose that it would be an unhelpful distraction to discuss these interactions in detail; hence, we have not followed the referee's recommendations to discuss these interactions in detail, beyond the extra details now added in lines 69-80.

3. *Following on from the above points, I am confused by the section on lines 266 to 272. It is not clear to me how the authors have deduced that there is no greater binding of chaotropes to hydrophobic regions than of kosmotropes. The m_L value does follow the hofmeister series suggesting that there is at least a difference in the solvent-mediated interactions, although it is not clear if this arises due to solvent-mediated interactions about polar groups or about non-polar groups?* In the revised version, the relevant lines are 300-305. Thank you for this point: the most important evidence comes from changes not to backbone ^{15}N and $^1\text{H}_\text{N}$ shifts, but to aliphatic ^{13}C and $^1\text{H}_\text{C}$ shifts, which are presented in Supplementary Material. We have therefore added a reference to supplementary material in the text. In line with the comments above, we have reworded the subsequent text to remove the element of "competition" between theories. On the final point: Yes, the m_L values do follow the Hofmeister series and thus imply a difference in solvent-mediated interactions. Interactions of solvent with polar and non-polar protein surfaces will clearly be different, but as noted in #2 above, because we place the emphasis more on the effect of co-solutes on fluctuations (the main ones being solvent-solvent, solvent-protein and protein-protein), then the detailed nature of the solvent-protein interaction surface becomes less important. Our emphasis in the text is on hydrogen bonding, because this is what protein HN chemical shifts tell us. These will mainly relate to changes around polar groups, but our data do not allow us to comment much on non-polar interactions. However, the general lack of shift change and curvature for non-polar groups does imply little direct interaction with non-polar groups.

4. *In Figure 9, I think it would be insightful to compare sulfate and chloride with either ectoine or betaine, and then to compare sulfate/chloride to SCN⁻. A strong correlation indicates that the solvent mediated interaction is only determined by the interaction of the co-solvent with water and independent of the chemical nature of the protein surface.* This is an interesting suggestion, which we have looked into. We comment in the text that the correlation is strongest when the co-solutes do not bind to the protein. The averaged binding affinities go in the order betaine < ectoine < TMAO ~ Cl⁻ < SCN⁻ < SO₄²⁻, implying that the best correlation is for betaine and ectoine (R=0.93), and the worst correlations are anything correlating to sulfate (all around R= 0.35). Other correlations have intermediate values (TMAO/ectoine 0.63, TMAO/Cl 0.65, ectoine/Cl 0.47, Cl/SCN 0.57). In other words, the quality of the correlation depends mainly on the extent of direct binding to the protein surface, which effectively introduces noise into the correlation. The data are unfortunately too noisy to provide information on the effect of the chemical nature of the surface.
5. *On line 502, I would suggest the following clarification.. : "any co-solute added to a protein will increase protein stability if the co-solute does not preferentially interact with the protein", note that denaturing co-solutes (eg urea) decrease protein stability because weak binding interactions overcome the volume exclusion effect.* Thank you, a useful suggestion, which we have included.
6. *On line 514, there is a mis-understanding about the meaning of fluctuations. KB integrals relate to density fluctuations of the solvent and co-solvent molecules, which are "static" properties, and do not relate to any dynamic property (eg folding rate constants, diffusion coefficients, conformational dynamics).* The referee is correct, and we have changed the text. We note that this raises the question of whether the ensemble average of a molecular system is the same as the time average – ie, whether the static fluctuation is equivalent to the time fluctuation. Yes it is, but only if the system is ergodic. It is generally agreed that molecular systems at equilibrium are ergodic, implying that very likely these two quantities are the same. But I certainly would not want to discuss that question here!
7. *Paragraph starting on line 523 - As far as I am aware, most approaches applying KB theory treat the protein as a rigid particle so it is not clear how protein fluctuations would relate to the solvent and co-solvent density fluctuations.* Another excellent point. The text relates solvent fluctuations to protein fluctuations via slaving, citing the classic Frauenfelder paper. That paper is explicitly about dynamic fluctuations rather than static fluctuations, but the fluctuations are linked via changes in free energy, so I feel we are on safe grounds in invoking an ergodic equivalence again to conclude that fluctuations at the solvent/protein interface will lead to fluctuations within the protein.

Reviewer 3

We thank the referee for their careful consideration.

1. *The authors explain all observed chemical shift perturbations exclusively by changes of hydrogen bonds. Generally, chemical shifts are influenced by various other factors and do not directly correlate with simple hydrogen bond strength alone. The authors should describe other possibilities or reasons why the authors eliminated them.* A good point. We have now provided more detail (lines 128-132). We measured a large variety of shift changes and tried hard to use them, but the only ones that were interpretable were for amide protons. This fits a consistent pattern discussed by us and many others, as described in the review now included as ref. 34, where amide protons (and to a minor extent amide nitrogens) are the only nuclei to give useful titration information.

2. *The authors introduce a new variable, m_L , to account for indirect effects leading to chemical shift perturbations. These indirect effects encompass changes in solvation other than intermolecular interactions, predominantly focusing on hydrogen bond strength. While the calculated m_L values correlate with hydrogen bond strength, the authors do not distinguish between hydrogen bonds formed with water molecules on the molecular surface and those within internal molecular atoms. This lack of distinction raises questions about the validity of treating these different types of hydrogen bonding equivalently. Specifically, amide protons on the molecular surface are in continual exchange with hydrogen atoms in water, in addition to forming hydrogen bonds. Considering the timescales of chemical shift observations and the hydrogen atom exchange rate, is it justifiable to overlook hydrogen atom exchange and other effects? Moreover, given that changes in solutes also impact chemical shift perturbations inside the molecule, it implies that the structure and dynamics of the entire molecule are affected to some extent. Is it accurate to attribute these changes solely to variations in hydrogen bond strength? Should the analysis also consider other factors influencing chemical shifts, such as ring-current effects? As discussed extensively in the text (in the final version, lines 306-330 and 534-582), we expected solvent-exposed amides and buried amides to have different m_L values, because exposed amides hydrogen bond to water while buried ones bond mainly to protein amide carbonyls, which we expected to change in different ways. We did not discuss the question raised here, whether hydrogen exchange between amide protons and water will have an effect. This is because it is well established to have no effect, mainly because the exchange rate under these conditions (pH 5.8) is very slow. In fact even at pH 7 or above, where exchange broadening starts to occur, chemical shift changes due to exchange remain unobservable. The referee is correct in saying that structure and dynamics (probably – see responses #6 and #7 to reviewer 2) of the protein are affected by addition of co-solutes. We and others (eg refs 32-34 in the final version) have shown that overwhelmingly, changes in amide proton shifts can be ascribed to changes in hydrogen bond geometries. There are certainly examples where significant changes to amide proton shifts are have been caused by ring current shifts, but these are very unusual (ref 34).*
3. *The authors assert that only a single ion binds to the protein surface, as indicated by K_d values, and that the fluctuation of this ion on the surface leads to chemical shift changes. This claim prompts a critical inquiry into the timescale of ion fluctuation on the surface. How does this timescale align with the timescale over which fast-exchange chemical shift changes are observed? The paper mentions that the results of ion fluctuation on the surface are replicated by MD simulations, etc. It would be beneficial for the authors to elucidate the correspondence of these timescales with those known from MD simulations, to reinforce the validity of their findings. Molecular dynamics calculations suggest contact times for chloride and sulfate with protein surfaces in water in the range of 0.1-0.8 ns (eg Vrbka et al J Phys Chem B 2006 110:7036-7043, doi 10.1021/jp0567624). This is a few hundred times longer than the typically quoted water residence time at a protein surface, and thus feels inherently reasonable. This is the total time for an ion to remain close to the protein surface before it dissociates, meaning that the time spent at any single binding site must be at least an order of magnitude shorter, ie less than 0.1 ns, or an exchange rate of at least 10^{10} s^{-1} . For slow chemical exchange to lead to noticeable signal broadening, the exchange rate needs to be within a factor of 100 of the chemical shift change, measured in Hz. A very generous estimate of the chemical shift change arising from binding of an anion would be 1 ppm, which on a 600 MHz NMR instrument means that the exchange rate must be as slow as 600*

$\times 100$ or $6 \times 10^4 \text{ s}^{-1}$. Thus, exchange is fast enough not to cause line broadening, by a factor of at least 10^5 . We have added a short comment to this effect into the text.

4. *The discussion section of the paper is extensive and complex that could hinder comprehension, especially due to its text-heavy format. Considering that Communications Chemistry is a general chemistry journal, not specifically focused on physical chemistry, and that its audience includes biochemistry researchers, it is crucial to present more concise and easily understandable explanations. The incorporation of several schematic diagrams would significantly aid in visually conveying the authors' concepts, thereby improving the paper's overall clarity. Additionally, the manuscript would be strengthened by the explicit inclusion of the physico-chemical equations that form the basis of the discussion, which are currently only described textually. Accompanying these equations with clear, explanatory notes would render the content more accessible and engaging for a wider readership.* Thank you for the excellent suggestion. We can see the value of an explanatory Figure that encapsulates the paper in a more easily understood way, and have included a new figure (10) along these lines. The reviewer seems to be suggesting that we should include Kirkwood-Buff equations accompanied by simple text describing how the equations can be applied in this context. I am afraid this is beyond our capability! KB equations are particularly opaque. Partly because of the nomenclature: there are lots of subscript indices, partial differentials and integrals, and the KB integrals are normally written using the symbol G , with changes in the integrals therefore written as ΔG , which I have always found very confusing because they have nothing to do with free energy. I have yet to find an "easy" account of KB theory. Therefore regrettably, we feel that inclusion of the equations will have exactly the opposite effect to that proposed by the reviewer.

Minor points:

1. *p3 L39: In the manuscript up to this point, there is a detailed explanation of kosmotropes and chaotropes. However, there is no definition for Hofmeister anions and osmolytes, and no explanation regarding their relationship between them, and these terms suddenly appear here. It is essential to include a clear and concise explanation of Hofmeister anions and osmolytes, particularly elucidating their relationship with kosmotropes and chaotropes, to avoid confusion and ensure clarity for readers not familiar with these terms.* We have added explanatory text at this point (lines 100-106).
2. *p3 L119: "while the linear parts represent indirect effects on the protein chemical shifts caused by the interactions of the co-solute with the solvent, which thus lead to altered protein solvation". It is mentioned that the linear chemical shift change in mL of the slope continues at least up to 1M. However, it seems implausible that this trend would continue indefinitely. It is reasonable to expect that the change will eventually reach a saturation point where the slope alters. The manuscript should address this aspect: up to what point can this linear relationship be considered valid? Clarification on the extent to which this theory can be applied is necessary to understand the limitations and applicability of these findings.* We have added text to clarify this (lines 155-157).
3. *p5 L164: "Almost the only generalisation that one can make about chemical shift changes is to say that a downfield shift of an amide proton indicates an increase in the strength of hydrogen bonding involving that proton." It is noted that the discussion up to this point has been based on the data presented in Figure 1. However, Figure 1 illustrates the chemical shift of ^{15}N . This raises a question: why have the authors chosen not to utilize ^1H chemical shifts in their analysis? The manuscript should provide an explanation or justification for the use of ^{15}N chemical shift data in the context of discussing hydrogen-related phenomena.* Chemical

shift changes in ^1H , ^{15}N and ^{13}C were all used for the calculation of affinity constants (eg Figs 2 and 4). For a structural interpretation of m_L , we only used ^1H for the reason given here. Fig 1 could equally well have used ^1H data; we used ^{15}N shifts partly to make this precise point, that one need not be limited to using ^1H data only.

4. *Equation (1): The variable 'i' is used but not clearly defined. Clearly specify it.* Done (line 166).
5. *Figure 2: The manuscript or figure should specify the criteria used for the color changes mapped onto the structure. Additionally, the text discusses certain amino acid residues (35-46, 81-86), but their locations are not clearly identifiable in the structure depicted in the figure. To aid reader comprehension, please either label the amino acid numbers directly on the structure, or number the secondary structures ($\alpha 1$, $\alpha 2$, $\beta 1$, etc.) and clearly indicate which regions are being discussed.* A useful point. We added text in the Methods section (lines 712-716) to indicate how the colors were generated. To avoid cluttering up an already crowded figure, we have added a labelled structure to the Supplementary Material (Figure S4).
6. *p7 L215: 'almost entirely linear.'. It is unclear which specific data or part of the analysis this statement refers to. For clarity and accuracy, it is essential that the manuscript explicitly specify the exact data or results being described as 'almost entirely linear.'* Table S1 presents the full set of fitted parameters. The criteria used to identify linear shift changes are listed in Methods. By applying these criteria to the data in Table S1, it is straightforward (although rather tedious) to identify the linear data. We have added text to indicate this (lines 246-247). If the reader wished to conduct a more comprehensive analysis, we would provide the data in the form of computer readable lists.
7. *p7 L225: "This region of the protein is not the most positively charged patch (which is the active site of this RNase), but it is close to the active site and to other positively charged regions.". The manuscript lacks a diagram of the electrostatic potential map on the 3D structure, making it difficult to identify charge locations. The manuscript should include such a diagram.* This reviewer's major point #4 included a recommendation for a new schematic / summary diagram, which we have added as Figure 10. This figure incorporates an electrostatic potential surface. Admittedly the surface only shows one face of barnase, but since this is the face containing the active site and the binding sites for sulfate and thiocyanate (and the other face is much less clearly delineated) we hope this satisfies the reviewer.
8. *p7 L235, "residues 27-31 and 54-62": Not sure which area in Figure 2c. Please indicate clearly in the Figure.* This region can now be identified using the residue numbering included in Figure S4. It is also identifiable as the region with the sulfate binding site shown in Figure 10.
9. *p8 L264: "Further support for this view comes from the fact that Hofmeister effects are widely held to be directly proportional to co-solute concentration, up to and beyond 1 M." References should be given.* Good point: we have done this.
10. *p8 L270: "We therefore rule out the preferential interaction theory, and also the preferential hydration theory, which leaves as possible explanations something like the original chaotrope/kosmotrope idea (ie for example that kosmotropes withdraw water from the protein surface) and the volume excluded theory, which has many elements in common with the chaotrope/kosmotrope idea." "preferential interaction theory", "preferential hydration theory", "The terms "preferential interaction theory", "preferential hydration theory", "original chaotrope/kosmotrope idea", etc. are not explained, and the point is not clear. It would be better to move these sentences to the discussion section and give more detailed explanations.* Thank you: a good point, and the text relevant to these points has already been changed and simplified in response to reviewer 2 #1.

11. p8 L277: *"A positive 1HN mL (as found for the kosmotrope sulfate) implies that titration with sulfate increases the strength of hydrogen bonding from the surface of the protein to water." For residues exhibiting negative mL values, which represent buried amide protons, discussing hydrogen bond with water seems incongruous. Could the authors clarify their explanation for this scenario?* Our apologies: the text here was not clear. We have added text to explain that this section is referring only to solvent exposed amides.
12. p9 L290: *"bulk water is more ordered in the presence of kosmotropes." Could the authors clarify how bulk water, in its ordered state, is related to the process of drawing hydration water from the protein surface?* We have added a reference to this statement, which originates from Hofmeister himself.
13. p9 L290: *"This would lead us to expect that buried amides would have mL with an opposite sign to the mL of surface-exposed amides. ." In Figure 3, there doesn't seem to be a distinct trend distinguishing positive mL values from negative ones. Could the authors specify the exact region of change they are referring to in this statement? Additionally, in the sentence above, the manuscript mentions, "proteins are very complex systems with dynamics". Yet, it concludes that chemical shift perturbations imply 'increasing the strength of interactions elsewhere (for example, inside the protein).' How can the authors assert that the interactions within the protein, presumably hydrogen bonds, have become stronger based solely on chemical shift perturbations?* We have added extra text (lines 323-324) to explain the first statement. The assertion is based on the statement (made several times in the paper and discussed extensively in these responses) that downfield changes in amide proton shifts indicate an increase in hydrogen bond strength and we feel there is no benefit in justifying this statement again here. We have however added a reference to support the statement in the paper that proteins "have the property that weakening of noncovalent interactions in one region ... is compensated for by increasing the strength of interactions elsewhere".
14. p15 L469: *"the binding of an ion, even when that binding is spread over a large surface patch, reduces dynamic fluctuations at the surface. " The logic of this and the subsequent statement is unclear. Based on the description provided, the binding of an ion to a protein surface seems to involve the ion moving around on the protein surface. How does this movement clearly suppress protein surface fluctuations? Is it possible to conclusively investigate such dynamics, for instance, through NMR relaxation analysis? Additionally, the manuscript should clarify the experimental basis for the assertion that suppressing protein fluctuations aids in the binding of a second ion. It is commonly believed that greater flexibility in a protein's binding surface allows for more diverse binding. Does this principle not apply to ions?* On re-reading our text, this is another place where we have conflated static fluctuations with dynamics (reviewer 2 #6). We have accordingly re-worded the text (lines 509-515) to make it clear that we are specifically discussing fluctuations (ie essentially an entropic effect) rather than dynamic motions. This goes a long way to addressing the concerns of the reviewer. As discussed in the context of responses to reviewer 2 #6 and #7, it is very likely that increased fluctuations will equate to increased dynamics, which we are currently investigating using NMR relaxation as suggested by the reviewer. The experimental basis for the assertion is the observation that binding affinities become stronger in the presence of other ions (lines 353-355, discussed here in lines 500-515). This is not an experimental observation specifically of reduced fluctuations, other than the point that ion binding is localised on the surface. This also addresses the final point raised by the reviewer: we always need to bear in mind that any interactions are always competing with water. It is the fluctuations of water that give rise to fluctuations in the protein (this is the basis of Frauenfelder's "slaving" argument). If the bound ion has fewer fluctuations (which it does,

because it is restricted to a specific binding region on the surface), then this will reduce the overall protein fluctuations. We have not added this point to the text, because it would complicate an already dense text.

15. *p16 L520: "KB theory emphasises Kirkwood-Buff integrals ..." Please describe the formula and explain it clearly. It is not clear from the text alone.* This is roughly the same point discussed above (reviewer 3 major point #4). We could write down the KB equations, which include the integrals. However, we do not feel that it would help to clarify the explanation. For the small number of readers who already understand KB theory, any text we write will not help; and for the majority to whom KB theory is mysterious, nothing that we can write here is likely to generate more clarity. See Ref 30, which is the clearest explanation I know, but is not exactly easy reading.
16. *p16 L520: "Lower solvent fluctuations reduce the ability of the solvent to hydrate the protein surface" I did not understand the logic behind this.* This statement is expanded in lines 575-601 and summarised in lines 653-668 and Figure 10: ultimately this can be seen as altering the balance between entropy (greater fluctuations) and enthalpy (a deeper energy landscape, and thus stronger bonds and better hydration, at the expense of a reduction in entropy).
17. *Method section should state what stable isotope labelling was done - 13C, 15N uniform label sample?* Thank you – done.
18. *Figures S2 and S3, which show profiles of chemical shift changes and fitting, are not cited anywhere in the text. Please ensure these figures are referenced in relevant sections. This will provide clarity and support for the claim 'no significant curvature' in the titration of osmolytes, helping the reader to understand and verify this assertion.* Thank you – now done. As a consequence of our responses to reviewers, the ordering of figures in supplementary material has been altered, so these are now actually Figures S1 and S2.
19. *In the introduction, the authors express an intent to identify effective excipients for protein stabilization. It would be beneficial for the biological readership if the manuscript concluded with brief guidance or recommendations on which excipients to choose based on the study's findings. This addition would provide practical value and aid in the application of the research.* Thank you for your vote of confidence! Actually what we expressed as our intent was to produce a more coherent framework, which we hope we have done. We have slightly expanded our "excipient" summary at lines 649-652 to set out what qualities of an excipient we feel are most usefully explored. We are not formulations scientists, and this work was done on a single protein at low concentration. We are currently widening the scope of our investigations to address some of these questions.

We apologise for the length of this response, which is necessitated by the length, detail and quality of the reviewers' comments.

REVIEWERS' COMMENTS:

Reviewer #1 (Remarks to the Author):

The authors have addressed all of my previous comments and clarified all of my questions. Hence I recommend publication of the manuscript.

I have only one minor remark: I inadvertently have copied a wrong doi in my original review, when I wrote "In a recent paper (10.1038/s41598-023-29901-5) this difference has been ascribed to curvature, but the authors of this study also discuss conformational changes in the SI."

I apologize for this mistake. The paper I intended to refer to is by Rogers et al:

<https://doi.org/10.1038/s41557-021-00805-z>. Although the reference does not study proteins, I feel that the conclusions seem very relevant to the present manuscript and the authors may want to discuss their conclusions in light of these earlier findings.

Reviewer #2 (Remarks to the Author):

As mentioned before, I think this article could be very ground breaking, my one concern though is that I find the introduction section still to be confusing, specifically the paragraph between lines 65 and 86. This section corresponds to a discussion in part comparing the work from Timasheff to the volume exclusion effect. There is a statement that the volume exclusion effect is based on a clear thermodynamic premise, but this is very misleading, as it implies that Timasheff's work is not based on thermodynamics. Rather Timasheff's work is grounded in thermodynamic linkage phenomena developed by Wyman in middle half of the 20th century. The volume exclusion effect is one mechanism for a co-solute (or co-solvent) to be excluded from around the protein, which can be interpreted in terms of preferential interaction parameters (measured by Timasheff) or in terms of Kirkwood-Buff integrals as alluded to in the current article. For instance, in Bhat and Timasheff (1992) *Protein Science* v1:1133, they related the preferential interaction measurements to the steric exclusion of polyethylene glycols from around protein surfaces. This is an example of a volume excluded effect, which can be quantified from direct thermodynamic measurements of the preferential interaction parameters. I would also like to draw the authors attention to an article by Smith (2006) *Biophys J* v91: 849, which provides just one example article where they show relationships between preferential interaction parameters (often denoted as dm_3/dm_2 where 3 denotes the co-solute or co-solvent and 2 the protein) and Kirkwood-buff integrals (see equation 16 of the paper). KB integrals are often applied when doing simulations since simulations provide access to the distribution functions G_{ij} . The important point is that to relate a simulation to a thermodynamic parameter, the most direct route is to relate the Kirkwood-Buff integrals to the preferential interaction parameter, which is the property measured by Timasheff. One other important point is that fluctuations with respect to intensive variables relate to second derivatives of free energy with respect to the corresponding intensive variable. So for instance, heat capacities relate to second derivatives of free energies. From a simulation, a heat capacity is related to the energy fluctuations, or the width of an energy histogram. Similarly density fluctuations relate to the width of a particle number histogram when a simulation is carried out in an open ensemble, these density fluctuations relate to second derivative of free energy with respect to co-solute concentration. For

instance see Equation 2 in Smith (2016), which relates the KB integrals to a derivative of chemical potential with respect to a species concentration. A chemical potential is itself a derivative of a free energy with respect to species concentration, so equation 2 shows how KB integrals relate to second derivatives of free energies. Indeed these relationships are used for showing how interactions between co-solvents and proteins relate to protein folding stability.

The implications of what I wrote above to the paragraph on page 2 is that the volume excluded effect is one molecular mechanism for how a co-solvent interacts with a protein surface, other mechanisms are that the exclusion of chaotropes/kosmotropes is related to their water-structure making or breaking ability, chaotropes adsorb to non-polar surfaces, etc... Note that preferential hydration could be driven by a volume exclusion effect (see the Bhat and Timasheff reference.). What determines how a co-solvent alters protein stability is determined by the net effect of all the different types of interactions, if the only mechanism is a volume excluded effect, this will indeed increase the stability since the exclusion about the unfolded state will be greater than the exclusion about the folded state. I would rephrase the last paragraph on page 2 to reflect the point that water structure breaking or making is one molecular mechanism for how a co-solvent interacts with a protein surface, while Kirkwood-Buff integrals are a way for relating interactions to thermodynamic properties, irrespective of the molecular mechanism for the interaction.

Reviewer #3 (Remarks to the Author):

The authors have answered most of my correction requests and questions, so I have no further comments anymore.

Dear Reviewers,

Thank you for your encouragement. In response to the referees' comments we have made a number of changes to the paper, highlighted in red (the first set of changes have now been incorporated into the text and are thus in black).

Reviewer #1

The authors may want to discuss their conclusions in light of [<https://doi.org/10.1038/s41557-021-00805-z>]. Thank you for pointing out this paper, which relates the binding of poorly hydrated anions such as thiocyanate to different curvatures of the molecular surface. It is difficult to relate this to protein binding, because the surface structures of proteins are complicated with a wide range of hydration and curvatures. However, this paper rationalises their observations by the very relevant point that thiocyanate binds preferentially to less well hydrated surfaces, essentially arguing that the displacement of bound water by thiocyanates is easier in regions where the water hydrogen bonding network is less structured. This point is very pertinent to our discussion, and we have therefore referenced and discussed it (line 77). Their analysis of NMR chemical shift changes has also followed very similar methodology to ours, and we have acknowledged their prior work (line 172).

Reviewer #2

We thank the reviewer for their feeling that this work could be ground breaking.

There is a statement that the volume exclusion effect is based on a clear thermodynamic premise, but this is very misleading, as it implies that Timasheff's work is not based on thermodynamics. We apologise for this lack of clarity, as this is absolutely not what we meant. We have therefore added the word *also* to indicate that we completely agree that Timasheff's work is based on thermodynamics (line 81).

The reviewer then provides a discussion/explanation of the difference between a mechanistic model (volume exclusion, for example) and a thermodynamic explanation (eg Kirkwood-Buff), referring to Smith (2006); and in the second paragraph, they expand this statement and recommend *I would rephrase the last paragraph on page 2 to reflect the point that water structure breaking or making is one molecular mechanism for how a co-solvent interacts with a protein surface, while Kirkwood-Buff integrals are a way for relating interactions to thermodynamic properties, irrespective of the molecular mechanism for the interaction.* Thank you, this is enormously helpful, and I have followed this recommendation. The changes made are on lines 88-92 and 546-547.

Reviewer #3

This reviewer raises no further points.

We thank the reviewers again for their helpful insights.

We have also made changes in line with the journal recommendations, which have necessitated numerous changes throughout and some increase in length. We have not identified these changes separately. We are attaching a version of the paper in its original format, with the changes made in response to the reviewers in red; and a separate version (for example with figures removed and journal recommendations implemented).